# Expression of nano-engineered RNA organelles in bacteria

Brian Ng [1,8], Catherine Fan[1,8], Milan Dordevic[1,8], Adam Knirsch [1,2,8], Layla Malouf [1], Giacomo Fabrini [3,4], Sabrina Pia Nuccio[3], Roger Rubio-Sánchez [1], Graham Christie[1], Masahiro Takinoue[5,6], Pietro Cicuta [2] & Lorenzo Di Michele [1,3,7] ✉

Designing synthetic biomolecular condensates, or membraneless organelles, offers insights into the functions of their natural counterparts and is equally valuable for cellular and metabolic engineering. Choosing *E. coli* for its biotechnological relevance, we deploy RNA nanotechnology to design and express non-natural membraneless organelles in vivo. The designer condensates assemble co-transcriptionally from branched RNA motifs interacting *via* base-pairing. Exploiting binding selectivity, we express orthogonal, non-mixing condensates, and by embedding a protein-binding aptamer, we achieve selective protein recruitment. Condensates can be made to dissolve and reassemble upon thermal cycling, thereby reversibly releasing and re-capturing protein clients. The synthetic organelles are expressed robustly across the cell population and remain stable despite enzymatic RNA processing. Compared with existing solutions based on peptide building blocks or repetitive RNA sequences, these nanostructured RNA motifs enable algorithmic control over interactions, affinity for clients, and condensate microstructure, opening further directions in synthetic biology and biotechnology.

Membraneless organelles (MLOs), emerging from condensation of proteins and nucleic acids, are found in most living cells, where they orchestrate biochemical reactions by sequestering and co-localising enzymes, substrates, and metabolites, and by maintaining homoeostasis[1–6]. Historically thought not to contain intracellular compartments, prokaryotes are now recognised to leverage membraneless organelles, such as BR-bodies, carboxysomes, polyP granules, ParB condensates, and RNA polymerase clusters, to modulate processes including division, polarisation, nucleoid replication, transcription, and protein homoeostasis[5–8].

Engineering non-natural, *designer* MLOs both in vitro and in vivo is key to unravelling the biophysics and functions of natural condensates[9–11], and equally valuable as a tool to program metabolism and cellular functions in synthetic biology and biomanufacturing[12,13]. To this end, establishing robust strategies to engineer biomolecular condensation in widely adopted prokaryotic chasses is particularly urgent.

Synthetic MLOs have been constructed from peptides and proteins, demonstrating remarkable functionalities[14–28]. Condensate-forming proteins, however, often assemble through non-specific interactions between Intrinsically Disordered Regions (IDRs), commonly combining electrostatic, π−π, and hydrophobic interactions between amino acid residues[29,30]. The complexity of these interactions makes them difficult to prescribe by design[31–33], with modelling often requiring advanced numerical simulations[34–36].

[1]Department of Chemical Engineering and Biotechnology, University of Cambridge, Cambridge, UK. [2]Cavendish Laboratory, Department of Physics, University of Cambridge, Cambridge, UK. [3]Department of Chemistry, Imperial College London, London, UK. [4]The Francis Crick Institute, London, UK. [5]Laboratory for Chemistry and Life Science, Institute of Integrated Research (IIR), Institute of Science Tokyo, Tokyo, Japan. [6]Research Center for Autonomous Systems Materiology (ASMat), Institute of Integrated Research (IIR), Institute of Science Tokyo, Tokyo, Japan. [7]fabriCELL, Imperial College London, London, UK. [8]These authors contributed equally: Brian Ng, Catherine Fan, Milan Dordevic, Adam Knirsch. ✉e-mail: ld389@cam.ac.uk

Alternative to proteins, naturally occurring RNA repeats can be deployed for MLO assembly[37–39], but their simple structure limits the scope for more advanced engineering.

Different from proteins, however, nucleic acids benefit from highly predictable base-pairing interactions, which have led to the emergence of robust, algorithmic strategies to design nanostructures and materials[40–44]. DNA nanotechnology has been successfully deployed to construct synthetic condensates with precisely controlled size and microstructure[45–50], and displaying advanced functions ranging from reversible assembly and disassembly[51–53], to large-scale morphological re-structuring[54–60], capture and release of functional payloads[51,61,62], and spatial organisation of biocatalytic pathways[63–66]. Despite this versatility, the use of synthetic DNA building blocks is limited to in vitro applications, as DNA nanostructures cannot be expressed by living cells[67].

To overcome this roadblock, here we deploy RNA nanostructures, which retain much of the programmability of DNA constructs[68–71], but can be synthesised in vivo[72] and in crowded environments that mimic cellular conditions[73]. We engineer *E. coli* to express branched RNA motifs that assemble into synthetic MLOs when featuring complementary base-pairing domains, while remaining largely soluble when designed to be non-interacting. Base-pairing selectivity enables the expression of orthogonal nanostructures, producing non-mixing, co-existing MLOs, while embedding aptamers facilitates specific protein capture. MLOs can be made to assemble and disassemble upon thermal cycling, reversibly releasing and re-capturing payloads in the process. Our results demonstrate the potential of co-transcriptional RNA nanotechnology for engineering non-natural MLOs, combining a robust and programmatic design paradigm with genetic encodability.

## Results

### Design and expression of condensate-forming RNA motifs in *E. coli*

The condensate-forming motifs consist of branched RNA *nanostars* (NS), with four double stranded (ds) 'arms' emanating from a central junction (Fig. 1A)[69]. Each nanostar arm terminates with a 9-nucleotide (nt) kissing loop (KL), consisting of a 6 nt recognition domain asymmetrically flanked by adenines (Fig. 1A). NS designs A and B feature palindromic, self-complementary domains in the KLs, producing affinity between same-type NSs and leading to condensation[69]. A and B KLs are mutually orthogonal, resulting in negligible cross-binding[69]. NS designs Ā and B̄ are identical to A and B, respectively, but have KL sequences scrambled to eliminate self-complementarity and prevent condensation[69]. Fluorescent Light-Up Aptamers (FLAPs) are embedded in one of the NS arms to facilitate detection and visualisation. Designs A and Ā include a 'Pepper aptamer' binding the HBC620 dye[74] (Fig. 1A(i) and Supplementary Fig. 1), while B and B̄ feature a 'Broccoli aptamer'[75] binding the TBI dye (Fig. 1A(ii) and Supplementary Fig. 1)[76]. All NS sequences are reported in Supplementary Table 1.

For all NS designs, correct expression, condensate formation (or lack thereof) and interaction orthogonality were confirmed with in vitro transcription experiments (Supplementary Figs. 2–8, Methods). Encouragingly, for in vivo expression robust condensation was observed in buffers mimicking the intracellular ionic conditions encountered in *E. coli* (200 mM K⁺ and 1 mM Mg²⁺[77,78]) (Supplementary Fig. 9).

NS sequences, followed by transcription termination domains[79,80], were then cloned into plasmids for expression in the *E. coli* BL21(DE3) strain with the IPTG-inducible T7 transcription system, as outlined in the Methods. Plasmid maps are reported in Supplementary Figs. 10, 11, while plasmid sequences are reported in Supplementary Data 1. *E. coli* were grown from a single colony in supplemented M9 medium to mid-log phase (OD 0.7) before inducing NS expression. Cells were characterised by confocal microscopy one hour post-induction (Methods).

We first tested *E. coli* expressing individual NSs: A, B, Ā or B̄. For both "sticky" motifs, A and B, visual inspection confirms MLO formation (Fig. 1B(i), C(i) and Supplementary Figs. 12, 13). Condensates appear to be overwhelmingly located at the poles of the cell, a common pattern in protein MLOs[8]. This observation is confirmed by averaging the fluorescence distributions of individual cells across the populations, producing the 2D maps in Fig. 1B(ii), C(ii). Pole localisation is likely to result from steric hindrance by the nucleoid located in the central section of the cell[81]. To test this hypothesis, in Supplementary Note 1, we use simple polymer physics arguments to estimate the energy cost of forming a small condensate (100 nm in diameter) within the nucleoid[82]. We find this penalty to be as high as $4.2 \times 10^{-20}$ J, or 10 $k_{\mathrm{B}}T$, which justifies the exclusion of the condensates from the central segment of the cells. In addition to the effect of the nucleoid, anisotropic confinement has also been hypothesised as contributing to pole accumulation of aggregates[8]. Pole accumulation consistently emerges from the distribution of MLOs positions that, however, also reveals a sparse third cluster in the central section of the cell (Supplementary Fig. 14). Image analysis confirms that most cells, 78% for A and 88% for B, contain two MLOs, while some feature three (20% for A and 8% for B, Fig. 1B(iii), C(iii)). Rarely, cells have a single MLO, and a negligible fraction is condensate-free. Detailed statistics on condensate numbers are reported in Table 1. RT-qPCR was used to quantify NS expression levels, yielding an intracellular concentration of $16 \pm 5\,\mu$M for NS B 1 hour post-induction (Supplementary Figs. 15, 16, Methods). By assuming that most NSs accumulate in the condensates, and approximating the per-NS volume in the condensed phase as that of a sphere with radius equal to the NS arm length (~8 nm[69]), we can roughly estimate the total condensate volume from the NS concentration. By further assuming that this volume is equally divided between two condensates, we estimate a mean condensate diameter of ~400 nm, compatible with microscopy observations (Fig. 1B(i), C(i)). Intra-cellular NS concentration is comparable with in vitro expression yields, measured as $13 \pm 4\,\mu$M 3 hours from starting the reaction (Supplementary Fig. 16).

Through image segmentation applied to single cells, we determined the integrated fluorescence intensity of individual condensates, $I_{\mathrm{cond}}$, expected to be proportional to their volume. Similarly, we computed the average intensity in the condensate-free region, $i_{\mathrm{disp}}$, proportional to the concentration of the dispersed NS phase, and the average intensity across the entire cell, $i_{\mathrm{tot}}$, proportional to the overall NS concentration (Methods). In Supplementary Fig. 17, we show that $i_{\mathrm{disp}}$ is weakly correlated with $I_{\mathrm{cond}}$ ($R^2 = 0.21$ and 0.24 for A and B, respectively), consistent with the notion that the NS behave similarly to an equilibrium phase-separating system, in which the concentration of the dispersed phase is independent on the volume of the dense phase. In addition, consistently with the lever rule, $i_{\mathrm{tot}}$ is strongly linearly correlated with $I_{\mathrm{cond}}$ ($R^2 = 0.70$ and 0.83 for A and B, respectively).

When identical staining and imaging settings are applied, typical fluorescence intensity recorded from condensates expressed in vivo closely matches the one measured in vitro, suggesting similar FLAP response and NS concentration in the condensed phase (Supplementary Fig. 18). Inspection of the time evolution of individual cells reveals occasional instances in which small condensates form in the central section, before merging with one of the polar MLOs, suggesting a pathway for condensate accretion alternative to single-monomer addition (Fig. 1D(i) and Supplementary Fig. 19A), as previously observed for repeat-RNA MLOs[38]. We propose that the rapid migration of mid-cell nascent condensates toward the poles is driven by the energetic cost associated with nucleoid deformation (Supplementary Note 1). These coalescence events hint at a liquid-like state of the MLOs. Consistent with the liquid-like state, Fluorescence Recovery After Photobleaching (FRAP) experiments revealed partial recovery (Supplementary Fig. 19B, C). It should however, be noted that recovery rates may be enhanced by the exchange of FLAP dyes[69].

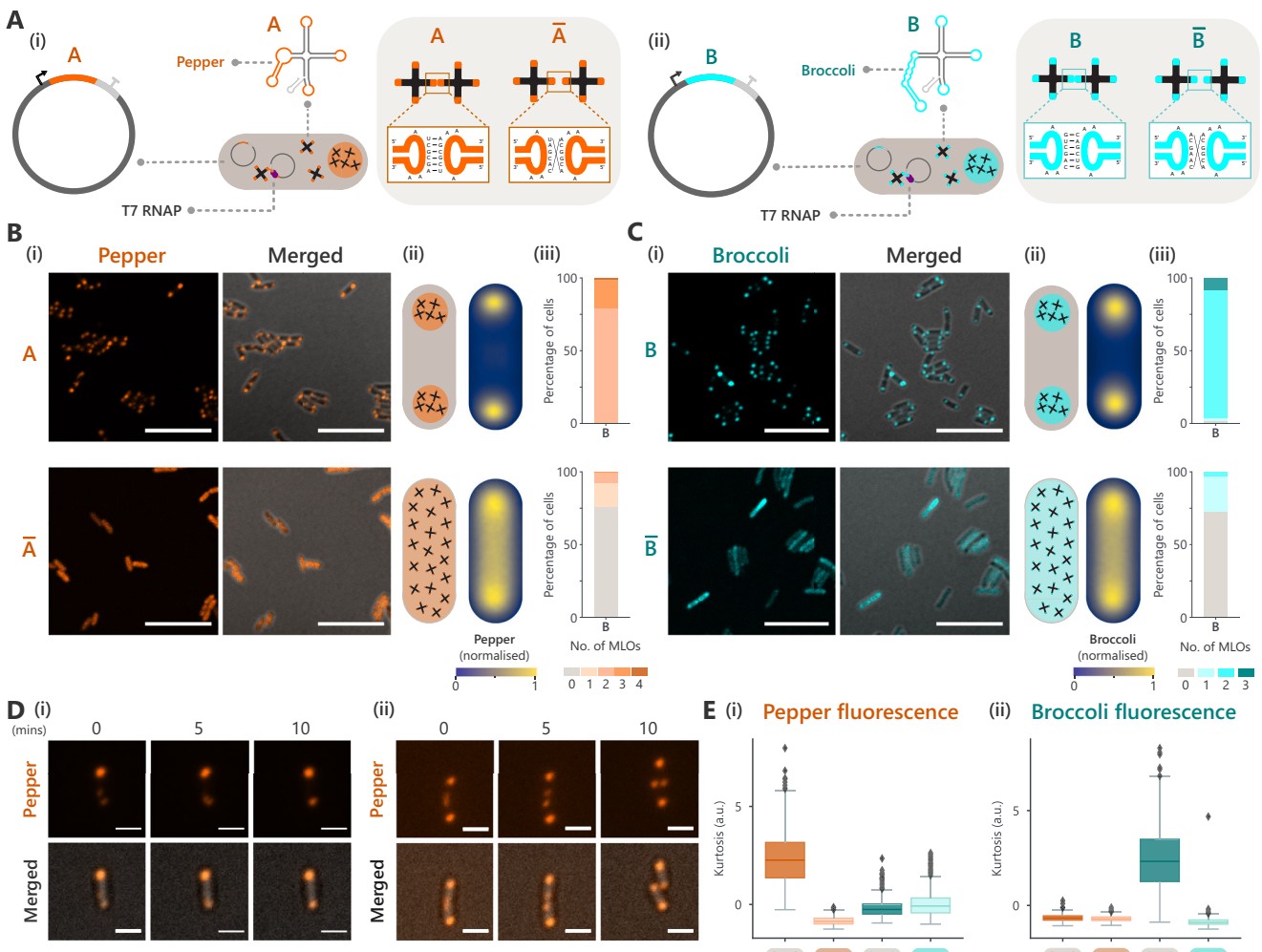

**Fig. 1 | Expression of nanostructured RNA MLOs in *E. coli*.** **A** Schematic of nanostar (NS) expression and membraneless organelle (MLO) formation in *E. coli*. (i) NS A contains four self-complementary kissing loops (KLs), driving condensation. NS Ā has non-complementary KLs with scrambled sequences, preventing assembly. Both A and Ā contain a Pepper aptamer, which binds to HBC620 activating its fluorescence[74]. (ii) NS B contains four self-complementary KLs, with a different sequence from that in A. B̄ has non-complementary loops and does not form MLOs. Both B and B̄ include a Broccoli aptamer[114], which binds TBI and activates its fluorescence[76]. Each NS sequence is cloned into a plasmid for expression with T7 RNA polymerase (T7 RNAP) upon transformation into *E. coli*. A terminator sequence (gray loop) is included. **B** Expression of A (top) and Ā (bottom) in *E. coli*. (i) Confocal micrographs. Left: Pepper fluorescence. Right: Merged Pepper fluorescence with bright-field. Scale bars, 10 μm. (ii) Schematic representation (left) and population-averaged maps (right) of Pepper fluorescence distribution in *E. coli* (Methods). (iii) Percentage of *E. coli* containing 0, 1, 2, 3 or 4 MLOs per cell. **C** Data as

for **B**, but for the expression of B (top) and B̄ (bottom) NSs in *E. coli*. **D** Time-lapse confocal micrographs of *E. coli* expressing A MLOs. Top: Pepper fluorescence. Bottom: Merged Pepper and bright-field fluorescence. Time is shown relative to the first frame (0 min). (i) MLOs fusion events within cells. Scale bars, 2 μm. (ii) Cell division in the presence of MLOs. Scale bars, 2 μm. **E** Box plots of the values of kurtosis extracted from the pixel-value distribution of fluorescence intensity in single cells. Analysis is performed on (i) Pepper and (ii) Broccoli fluorescence for *E. coli* expressing A, Ā, B and B̄ NSs (from left to right). The centre line and the lower and upper edges of each box represent the median and the 25th (Q1) and 75th (Q3) percentiles, respectively. Whiskers indicate the minimum and maximum values within 1.5 × IQR, while any points beyond this range are plotted as outliers (shown as diamonds). Outcomes of statistical tests are provided in Supplementary Tables 3 and 4. In **B**, **C** and **E**, data were obtained from 3 separate cultures, each starting from a single colony. Source data are provided as a Source Data file.

Time lapses additionally suggest that a third MLO may appear near the equator of cells with two polar condensates as they approach division, when the central section is free from the nucleoid (Fig. 1D(ii) and Supplementary Fig. 20). Consistent with this picture, for both A and B designs, we find a significant correlation between the length of the cells and the number of hosted MLOs (Supplementary Fig. 21). In some instances, the central condensates then appear to split between the two daughter cells, which emerge with two polar condensates of asymmetric intensity (Fig. 1D(ii) and Supplementary Fig. 20(i)). Over time, smaller condensates grow towards parity with the larger ones, progressively reducing size asymmetry (Supplementary Fig. 22). This evidence excludes Ostwald ripening as a dominant mechanism for condensate coarsening, consistent with observations made on DNA

nanostar condensates in vitro[83]. In other cases, the newly formed MLO partitions entirely in one of the daughter cells, leaving the other with a single condensate (Supplementary Fig. 20(ii)). Evidence of division confirms the viability of the MLO-expressing cells.

Expressing NS designs with scrambled KLs, Ā and B̄, largely results in uniformly distributed fluorescence (Fig. 1B(i)-(ii), C(i)-(ii) and Supplementary Figs. 23, 24), with a small fraction of the cells (23% for Ā and 28% for B̄, Fig. 1B(iii)-C(iii)) displaying regions of increased signal. These clusters likely arise from non-sequence-specific interactions between NSs or with other cellular RNA or proteins[84,85]. Average fluorescence maps (Fig. 1B(ii)-C(ii)) and location analysis (Supplementary Fig. 14) show that, like specific MLOs, non-specific NS clusters are preferentially located at the cell poles.

**Table 1 | Statistical snapshot of the number and percentage of cells expressing a given number of MLOs of each type**

| Dataset | Pepper channel (NSs A or Ā) | | | | | | Broccoli channel (NSs B or B̄) | | | | | n |
|---|---|---|---|---|---|---|---|---|---|---|---|---|
| No. of MLOs | 0 | 1 | 2 | 3 | 4 | Mean | 0 | 1 | 2 | 3 | Mean | |
| A | 1 | 3 | 379 | 98 | 4 | 2.21 | 485 | - | - | - | 0.00 | 485 |
|  | 0% | 1% | 78% | 20% | 1% | | 100% | - | - | - | | |
| Ā | 315 | 68 | 28 | 2 | - | 0.31 | 413 | - | - | - | 0.00 | 413 |
|  | 76% | 16% | 7% | 0% | | | 100% | - | - | - | | |
| B | 482 | - | - | - | - | 0.00 | 7 | 12 | 423 | 40 | 2.03 | 482 |
|  | 100% | - | | | | | 1% | 2% | 88% | 8% | | |
| B̄ | 505 | - | - | - | - | 0.00 | 366 | 124 | 15 | - | 0.30 | 505 |
|  | 100% | - | | | | | 72% | 25% | 3% | | | |
| AB | 3 | 327 | 72 | 1 | - | 1.17 | 33 | 277 | 92 | - | 1.15 | 402 |
|  | 1% | 81% | 18% | | | | 8% | 69% | 23% | | | |
| AB̄ | 1 | 128 | 283 | 1 | - | 1.69 | 301 | 109 | 3 | - | 0.28 | 413 |
|  | 0% | 31% | 69% | 0% | | | 73% | 26% | 1% | | | |
| ĀB | 257 | 23 | - | - | - | 0.08 | 9 | 193 | 78 | - | 1.25 | 280 |
|  | 92% | 8% | | | | | 3% | 69% | 28% | | | |
| ĀB̄ | 380 | 21 | - | - | - | 0.05 | 394 | 7 | - | - | 0.02 | 401 |
|  | 95% | 5% | | | | | 98% | 2% | | | | |

"Dataset" indicates experiment types with *E. coli* expressing one or a combination of NSs, as indicated. "Mean" indicates the mean number of MLOs per cell in each dataset. *n* is the number of cells analysed for each dataset. For each dataset, data are collated from 3 separate cultures, each starting from a single colony.

Owing to imaging noise and cell-to-cell variation, it is sometimes challenging to distinguish slight non-uniformities in the fluorescent signal from NS condensates, which may lead to false-positive MLO detections for Ā and B̄ (Methods). To discriminate more reliably, we analyse the frequency distribution of pixel values within single cells and extract values for the kurtosis (Methods). A positive kurtosis reflects highly localised fluorescence, characteristic of MLO formation, whereas a negative kurtosis reflects a more uniform fluorescence distribution (Supplementary Fig. 25). Consistently, we observe positive kurtosis for NS A and B, and negative values for Ā and B̄ (Fig. 1E, and Supplementary Tables 3, 4). Alternative to the kurtosis, fluorescence-distribution skewness also discriminates well between MLO-forming and scrambled NS designs (Supplementary Figs. 25, 26).

Optical density measurements performed in bulk cultures reveal an expected decrease in growth rate upon induction, which is marginally more prominent in *E. coli* expressing MLO-forming NSs compared to those producing non-sticky designs (Supplementary Figs. 27, 28A). Simultaneous detection of NS fluorescence shows the anticipated signal growth upon induction, with stronger emission observed for A and B NSs compared to Ā and B̄ (Supplementary Figs. 28B). Uninduced controls do not show significant induction leakage.

Given the limited half-life of mRNA in *E. coli*[86], it is expected that nanostars may also be progressively degraded. We thus investigated stability and processing of the expressed A and B NSs with nanopore RNA sequencing and Polyacrylamide Gel Electrophoresis (PAGE) (Supplementary Figs. 29, 30). We find that both constructs are cleaved at specific sites, located on two of the arms of NS A and on one arm of NS B. For both nanostructures, fragments corresponding to all sections of the designed transcript are detected, implying that the observed fragmentation is likely the result of endonucleases acting on folded constructs, rather than abortive transcription. In this scenario, cleavage of one of the dsRNA arms would reduce NS valency without impacting the folded construct downstream of the restriction site. This picture is consistent with the evidence of robust MLO formation: while a fraction of the NS population may have reduced valency, a sufficient proportion of the constructs must retain three or four KLs to support phase separation. Despite occurring at two distinct sites, enzymatic degradation is less pronounced for NS A compared to NS B (Supplementary Figs. 29B, 30B). Denaturing PAGE (Supplementary Figs. 29C, 30C) shows substantial quantities of the full, uncleaved transcript for NS A, which are not detected for NS B. Native PAGE shows the full-length construct as the dominant species for NS A, indicating that most cleavage events detected by sequencing may only nick the dsRNA arms, rather than fully detaching them, allowing the construct to retain the designed secondary structure (Supplementary Fig. 29D). For NS B, a moderate amount of the complete, folded construct is present (Supplementary Fig. 30D), likely featuring a nicked but not fully cleaved arm. The most abundant species, however, has lower molecular weight and likely corresponds to the NS lacking one KL and having, therefore, reduced valency (Supplementary Fig. 30D). This observation highlights the advantage of using tetravalent rather than trivalent NS designs: given that a minimum valency of three is required to sustain phase separation[69], the built-in redundancy of tetravalent constructs makes them intrinsically more resilient. In both NS designs, cleavage was not detected on NS arms that include the FLAP or the T7 terminator, possibly due to these structural features hindering RNase access[87]. Both NSs, however, feature a cleavage site in the terminator loop itself, which is thus likely to be detached in many transcripts (Supplementary Figs. 29A, B, 30A, B). The apparent protective effect of altering the duplex structure of NS arms could be leveraged to design against degradation. In addition, stability may be enhanced by circularising the constructs to prevent digestion by exonucleases[88].

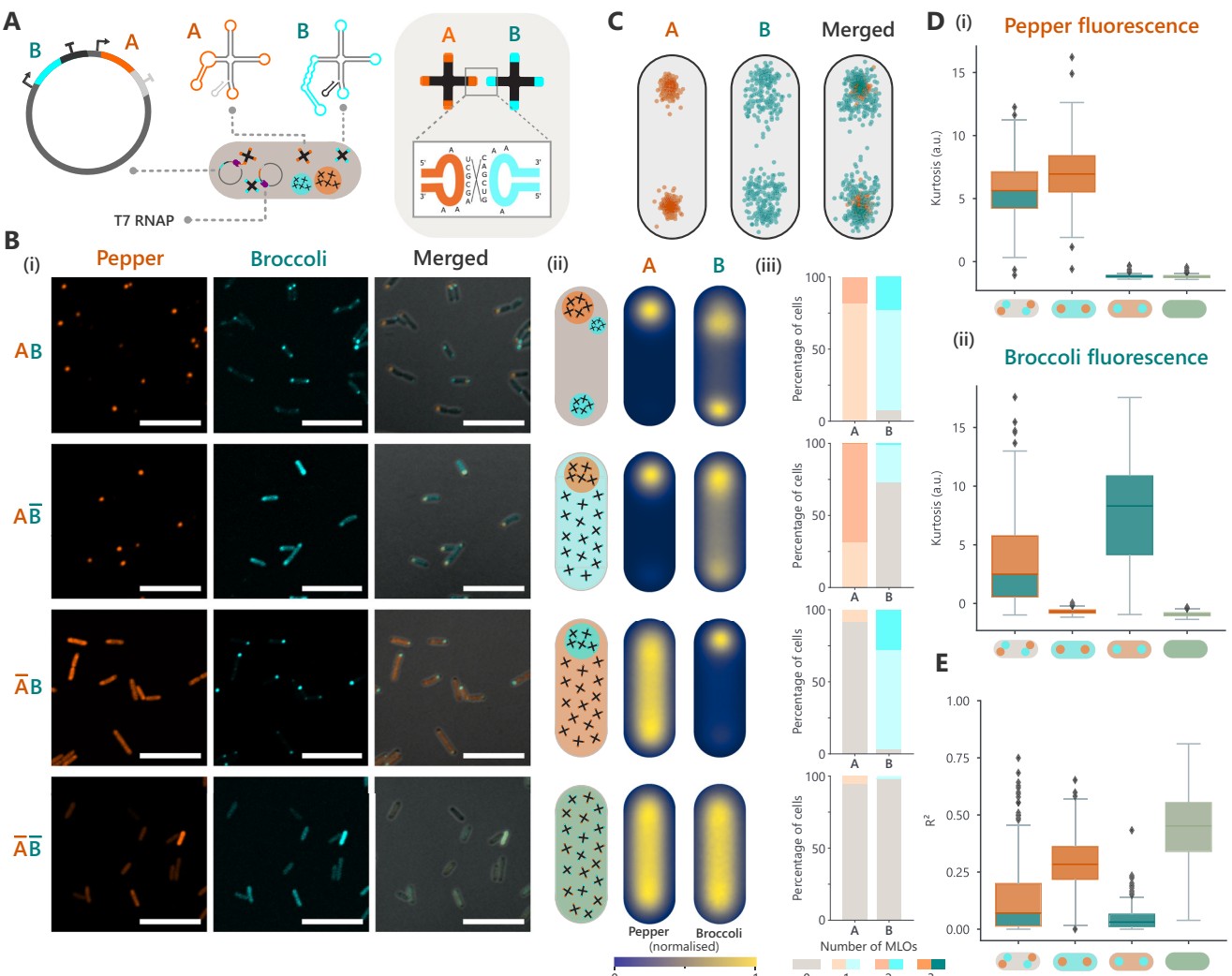

**Fig. 2 | Co-expression of orthogonal, non-mixing RNA MLOs in *E. coli*.**
**A** Schematic of the co-expression of A and B NSs in *E. coli*. A and B motifs have different kissing loop sequences and cannot interact by base-pairing, therefore forming separate MLOs in *E. coli*. Both NS sequences are cloned into a single plasmid for T7 RNA polymerase transcription, each controlled by its own promoter and a distinct terminator[80]. **B** Co-expression of NS pairs in *E. coli*. Top to bottom: A +B, A+B̄, Ā+B and Ā+B̄ NSs. (i) Confocal micrographs. Left: Pepper fluorescence (NSs A and Ā). Middle: Broccoli fluorescence (NSs B and B̄). Right: Merged Pepper fluorescence, Broccoli fluorescence, and bright-field. Scale bars: 10 μm. (ii) Schematic representation (left) and population-averaged maps of Pepper fluorescence distribution (middle), and Broccoli fluorescence distribution (right) in *E. coli* (Methods). (iii) Percentage of *E. coli* containing 0, 1, 2 or 3 A or B MLOs per cell. **C** Scatter plot of the locations of MLOs in *E. coli* co-expressing both A and B NSs.

Each symbol represents the position of a detected MLO (Methods). **D** Box plots of the values of kurtosis extracted from the pixel-value distribution of fluorescence intensity in single cells. Analysis is performed on (i) Pepper and (ii) Broccoli fluorescence for cells expressing A+B, A+ B̄, Ā+B and Ā+B̄ NSs (left to right). Outcomes of statistical tests are provided in Supplementary Tables 5 and 6. **E** Box plots of the correlation coefficient ($R^2$) values computed between Broccoli and Pepper fluorescent signals. Outcomes of statistical tests are provided in Supplementary Table 2. In **D**, **E**, each data point refers to an individual *E. coli*. The centre line and the lower and upper edges of each box represent the median and the 25th (Q1) and 75th (Q3) percentiles, respectively. Whiskers indicate the minimum and maximum values within 1.5 × IQR, while any points beyond this range are plotted as outliers (shown as diamonds). In **B**, **E**, data are obtained from 3 separate cultures, each starting from a single colony. Source data are provided as a Source Data file.

## Expression of orthogonal membraneless organelles in *E. coli*

KL orthogonality between A and B designs should enable co-expression of non-mixing MLOs. We test this hypothesis by designing plasmids containing two distinct NS sequences (Fig. 2A and Supplementary Fig. 11). Different T7 terminator sequences[80] are used for the two NSs to minimise the potential for non-specific affinity between the constructs induced by the terminator loops.

When co-expressing A and B, confocal microscopy confirms the formation of non-mixing MLOs (Fig. 2B(i), Supplementary Figs. 31, 32). Similar to single-NS systems, condensates are preferentially located towards the poles, as evident from both average fluorescence maps (Fig. 2B(ii), Supplementary Fig. 33) and the locations of individual MLOs (Fig. 2C). Of the cells containing one A-type and B-type MLO

(60% of all cells co-expressing NSs A and B), 66% had both MLOs positioned at the same pole. This asymmetric distribution may stem from the cells' division history, as newly formed poles are less likely to harbour condensates. Alternatively, or in addition, it may be driven by weak, non-specific affinity between A and B condensates. We also observe a clear asymmetry between the exact positioning of A and B condensates: the former are nearly always in contact with a pole, while the latter may occupy more central locations (Supplementary Fig. 14). Inspection of individual micrographs confirms that B condensates tend to accumulate around polar A MLOs (Supplementary Fig. 31). We ascribe this behaviour to the tendency of A NSs to form larger condensates (Supplementary Fig. 34), which may appear faster than B MLOs and grow to occupy the entire polar region, pushing smaller B

condensates more towards the central segment of the cell. Slower condensation kinetics for the B design may be linked to its greater susceptibility to degradation and consequent reduced valency.

Most cells express one MLO of each kind, with a smaller percentage containing two A and/or two B MLOs (Fig. 2B(iii) and Table 1). The overall number of MLOs in cells co-expressing A and B (averaging 2.3) is similar to that of cells expressing only A (2.2) or only B (2.0), indicating that co-expression does not significantly alter condensate formation.

Cells co-expressing one sticky NS and one non-sticky NS, A + $\bar{B}$ or $\bar{A}$ + B, behave as expected, with MLOs of the intended NS species (A or B) coexisting with a largely diffuse phase of the soluble NSs ($\bar{B}$ or $\bar{A}$, Fig. 2B(i) and Supplementary Figs. 35, 36). The fluorescent signal distribution (Fig. 2B(ii)) and condensate locations (Supplementary Fig. 14) of A and B MLOs are similar to those observed for cells that do not simultaneously express $\bar{B}$ or $\bar{A}$ (Fig. 1B(ii), C(ii) and Supplementary Fig. 14), indicating that soluble NSs do not interfere with the condensation of sticky motifs. Condensate numbers are, however, lower, with a greater proportion of cells expressing a single A or B MLO (Fig. 2B(iii)), arguably due to a lower overall expression of the MLO-forming motif due to the simultaneous production of the soluble NSs.

The fluorescence distribution of $\bar{B}$, when co-expressed with A, is largely uniform, with a small fraction of the cells (27%) displaying regions of increased $\bar{B}$ signal at the poles (Fig. 2B(ii)-(iii) and Table 1), similar to that observed in cells expressing $\bar{B}$ only (28%). The fluorescence distribution of $\bar{A}$, when co-expressed with B, is almost completely uniform, with only 8% of cells displaying regions of increased $\bar{A}$ signal (Fig. 2B(ii) and Table 1), compared to 25% for E. coli expressing only $\bar{A}$. Cells co-expressing the two soluble NSs, $\bar{A}$ and $\bar{B}$, display near-uniform fluorescence distribution (Fig. 2B(ii) and Supplementary Fig. 37), with non-specific $\bar{A}$ and $\bar{B}$ clusters detected only in 5% and 2% of the cells, respectively (Fig. 2B(iii) and Table 1).

The lesser tendency of $\bar{A}$ and $\bar{B}$ to form non-specific clusters is consistent with a reduced NS concentration caused by the simultaneous expression of two constructs. The greater abundance of non-specific clusters seen when non-sticky NS are co-expressed with sticky motifs, particularly for A+$\bar{B}$ expression, may be due to MLO-templated clustering of the otherwise soluble NSs. The soluble NSs may indeed weakly partition within, or coat the specifically-assembled MLO, or the condensates may serve as heterogeneous nucleation sites for non-specific aggregation. Either process may be facilitated by weak affinity between the soluble and sticky NSs due to base pairing, misfolding, enzymatic degradation, or the action of native RNA-binding proteins. This interpretation is supported by the evidence of weak partitioning of soluble NS $\bar{B}$ in NS A condensates observed in vitro (Supplementary Fig. 2), and quantified in Supplementary Fig. 8 (Methods).

As done for single-NS systems, we use the kurtosis of pixel-intensity distributions to assess the tendency of individual NS components to condense. In line with observations made on Fig. 2B, for all combinations tested, cell populations expressing A or B NSs have positive kurtosis in the relevant fluorescent channels, while those expressing $\bar{A}$ or $\bar{B}$ have negative kurtosis (Fig. 2D and Supplementary Tables 5, 6). Skewness analysis can similarly distinguish between cells expressing condensate-forming and non-sticky NSs (Supplementary Fig. 26).

To further assess the relative distribution of co-expressed NS components, we study the cross-correlation between Pepper (NS A) and Broccoli (NS B) fluorescence signals (Supplementary Fig. 38A), extracting distributions of the coefficient of determination, $R^2$, across E. coli populations (Fig. 2E and Supplementary Table 2). In cells expressing A and B, the low $R^2$ confirms that MLOs are orthogonal and non-overlapping. When expressing $\bar{A}$ and $\bar{B}$, $R^2$ is highest, as both NSs are evenly distributed throughout the cell. In E. coli producing $\bar{A}$ and B, a low $R^2$ follows from the localised Broccoli signal in a diffused Pepper background. The higher $R^2$ value found for cells expressing A+$\bar{B}$

supports the hypothesis that $\bar{B}$ may non-specifically partition within MLOs, consistent with observations made in vitro (Supplementary Figs. 2B, 8).

Bulk growth curves collected for E. coli expressing two NS species are similar to those recorded for single-NS expression (Supplementary Figs. 39A). Samples show no significant differences in the trend and magnitude of the fluorescent traces recorded simultaneously (Supplementary Figs. 39B). We also observe that some cells still contain MLOs after a week at room temperature, indicating long-term stability of the condensates despite enzymatic degradation of individual NSs (Supplementary Fig. 40).

## Selective protein recruitment in RNA membraneless organelles

Owing to their modularity and controlled structure, the RNA motifs can be straightforwardly modified to feature additional functional motifs, including aptamers enabling the capture of molecular clients. As proof-of-concept, we embedded the GFP-binding AP3 aptamer[89] in one of the arms of NS A, opposite to the arm hosting the Pepper FLAP (Fig. 3A). Note that the resulting $A_{AP3}$ NS remains tetravalent, as the kissing loop is inserted between two stem loops of the aptamer.

Confocal micrographs and image analysis confirm that $A_{AP3}$ expression in E. coli produces MLOs with similar spatial distribution and numbers as those measured for NS A (Fig. 3B(i)–(iii), Table 2, and Supplementary Fig. 41). Co-expression of GFP results in a clear co-localisation of the protein with the $A_{AP3}$ signal (Fig. 4B(i), (ii) and Supplementary Fig. 42). Cells expressing $A_{AP3}$ alone and those co-expressing $A_{AP3}$ and GFP both contained an average of 2.0 condensates per cell (Table 2), indicating that GFP expression does not noticeably affect $A_{AP3}$ condensate formation.

In contrast, GFP remains evenly distributed when co-expressed with NS A lacking the GFP-binding aptamer, confirming the specificity of the recruitment mechanism (Fig. 3B(i), (ii) and Supplementary Fig. 43). Expressing GFP in the absence of NSs confirms the lack of spontaneous protein aggregation and the formation of inclusion bodies (Fig. 3B(i), (ii) and Supplementary Fig. 44)[90]. The small number of GFP clusters detected by our segmentation algorithm in the absence of $A_{AP3}$ NSs come from segmentation artefacts linked to the 3D shape of the cell[91], which results in brighter central sections that may be occasionally misclassified as MLOs.

Partial recovery of the GFP signal in FRAP experiments demonstrates that the recruited protein can exchange with cytosolic GFP (Fig. 3C).

Analysis of the kurtosis of the pixel-fluorescence distribution in Fig. 3D shows the expected trends: positive kurtosis is measured in the Pepper (NS) channel for all MLO-forming NS designs, and in the GFP channel only when the protein-binding $A_{AP3}$ is expressed (Supplementary Tables 7, 8). Consistent trends are observed when analysing skewness (Supplementary Fig. 26). Finally, image correlation analysis shows a strong fluorescence co-localisation between GFP and $A_{AP3}$ MLOs but not between GFP and A MLOs (Fig. 3E, Supplementary Fig. 38, and Supplementary Table 2). Incidentally, the strong co-localisation between GFP and $A_{AP3}$ confirms the limited impact of potential chromatic aberrations between the red fluorescence channel (used for Pepper-labelled NSs) and the green channel (used for GFP and Broccoli-labelled NSs.)

## Temperature-dependent dissolution and re-assembly of MLOs

Since the NSs self-assemble through base-pairing interactions, we anticipated that the MLOs would dissolve at sufficiently high temperature, as observed in vitro (Supplementary Figs. 45–47). Consistent with this expectation, heating E. coli containing A MLOs led to their dissolution at 61 °C. This process was reversible, with A MLOs reassembling with a similar polar distribution as the cells were cooled (Fig. 4A(i)-(ii) and Supplementary Fig. 48). Sampling the skewness of the pixel-intensity distribution across the bacteria population provides

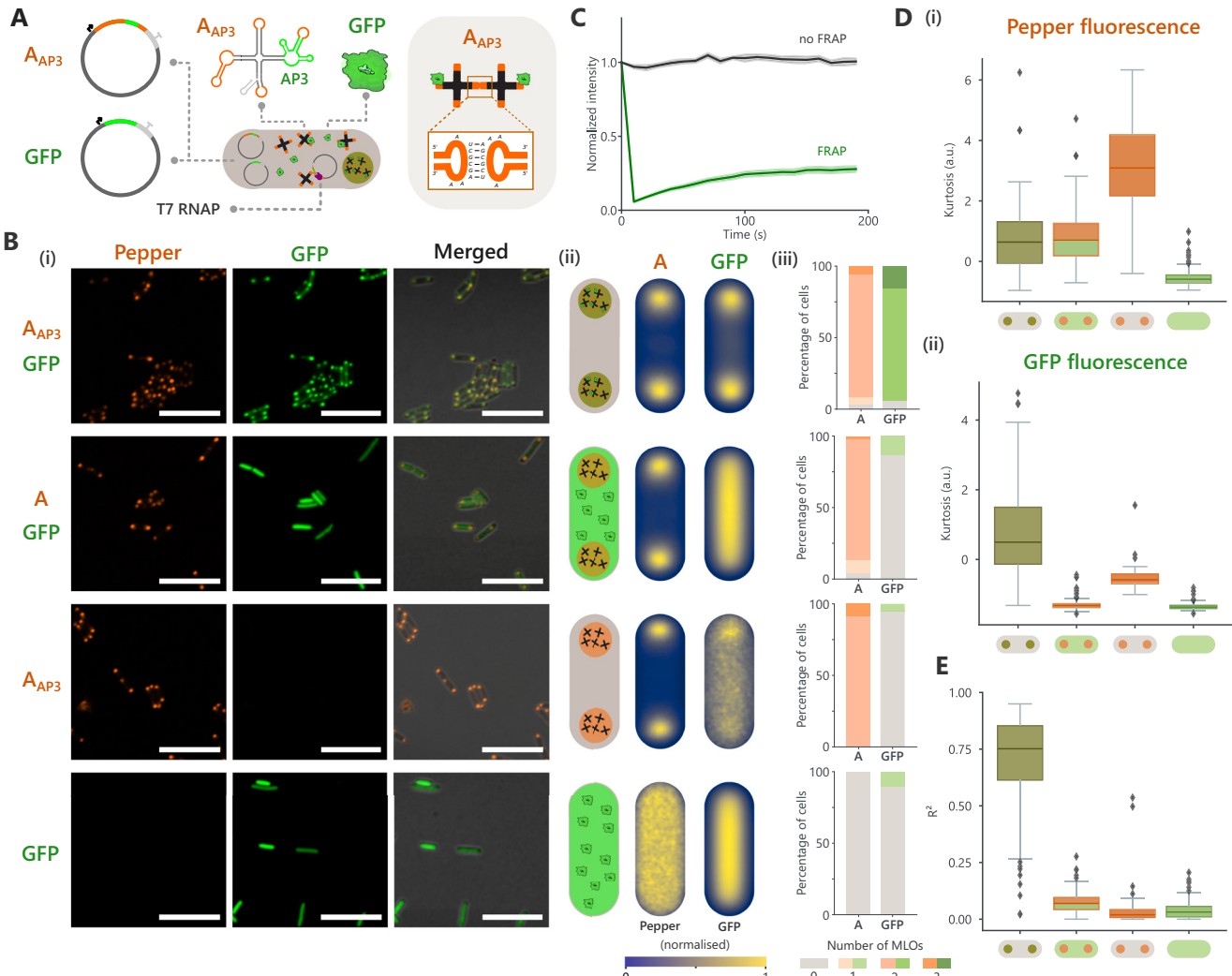

**Fig. 3 | Proteins can be selectively captured in RNA MLOs expressed in *E. coli*.**
**A** Schematic of GFP co-expression with NS A$_{AP3}$, carrying the GFP-binding AP3 aptamer[89], in *E. coli*. **B** Co-expression of NSs and GFP in *E. coli*. Top to bottom: A$_{AP3}$+GFP, A+GFP, A$_{AP3}$ only, and GFP only. (i) Confocal micrographs. Left: Pepper (NS) fluorescence. Middle: GFP fluorescence. Right: Merged Pepper fluorescence, GFP fluorescence, and bright-field. Scale bars, 10 μm. (ii) Schematic representation (left) and population-averaged maps of Pepper fluorescence distribution (middle), and GFP fluorescence distribution (right). (iii) Percentage of *E. coli* containing 0, 1, 2 or 3 MLOs per cell. **C** Fluorescence Recovery After Photobleaching (FRAP) curves of GFP captured in A$_{AP3}$ MLOs. For each MLO, fluorescence intensity prior to photo-bleaching is normalised to 1. The green solid line is the mean intensity of photo-bleached samples ($n$ = 25 condensates), while the black solid line is the mean intensity of non-photobleached samples ($n$ = 15 condensates). Shaded areas indicate the standard error of the mean. **D** Box plots of the values of kurtosis extracted

from the pixel-value distribution of fluorescence intensity in single cells. Analysis is performed on (i) Pepper and (ii) GFP fluorescence for cells expressing A$_{AP3}$+GFP, A +GFP, A$_{AP3}$ only, and GFP only. Outcomes of statistical tests are provided in Supplementary Tables 7, 8. **E** Box plots of the coefficient of determination ($R^2$) values computed between Pepper and GFP fluorescent signals. Outcomes of statistical tests are provided in Table 2. In **D**, **E**, each data point refers to an individual *E. coli*. The centre line and the lower and upper edges of each box represent the median and the 25th (Q1) and 75th (Q3) percentiles, respectively. Whiskers indicate the minimum and maximum values within 1.5 × IQR, while any points beyond this range are plotted as outliers (shown as diamonds). In **B-E**, data are obtained from 3 separate cultures, each starting from a single colony. Source data are provided as a Source Data file. GFP illustration was created with BioRender (https://BioRender. com/yeblcvk) and licensed under CC BY 4.0.

information on the condensate melting process (Fig. 4A(iii)). At the starting temperature, the skewness is high across the population, reflecting the presence of MLOs. As the temperature is increased, the distribution becomes bimodal, with a second population of cells displaying low skewness following MLO dissolution. The intra-population variability in the onset of melting may be due to cell-to-cell variation in NS expression and, therefore, concentration. At sufficiently high temperature, most cells display low skewness. By fitting the skewness distributions, we can extract a temperature-dependent fraction of cells hosting MLOs, which shows a small hysteresis between heating and cooling ramps. This may be caused by slow assembly kinetics, delays in thermal equilibration, or changes in protein crowding due to protein

denaturation during heating, which could alter the local environment of the MLOs (Fig. 4A(iv) and Supplementary Fig. 49). B MLOs showed a reversible disassembly behaviour similar to the A design, but with a lower melting temperature: condensates begin to melt at 49 °C and fully dissolve by 58 °C (Fig. 4B(i)-(iv) and Supplementary Figs. 49, 50). Single-cell skewness-temperature curves were used to estimate the melting temperature for each cell. Linear regression between the melting temperature and cellular fluorescence gave low coefficients of determination ($R^2$ < 0.2), indicating no detectable correlation between NS concentration and melting temperature (Supplementary Fig. 51). The difference in melting temperature between the two constructs is unlikely to result from different KL affinity, having observed that NSs

**Table 2 | Statistical snapshot of the number and percentage of cells expressing a given number of MLOs for protein capture experiments**

| Dataset | Pepper channel (NSs A or A$_{AP3}$) | | | | | GFP channel | | | | | n |
|---|---|---|---|---|---|---|---|---|---|---|---|
| No. of MLOs | 0 | 1 | 2 | 3 | Mean | 0 | 1 | 2 | 3 | Mean | |
| A$_{AP3}$+GFP | 5 (3%) | 7 (5%) | 125 (86%) | 8 (6%) | 1.94 | 9 (6%) | - | 114 (79%) | 22 (15%) | 2.03 | 145 |
| A+GFP | 9 (4%) | 18 (9%) | 174 (85%) | 4 (2%) | 1.84 | 177 (86%) | 28 (14%) | - | - | 0.14 | 205 |
| A$_{AP3}$ | - | - | 75 (91%) | 7 (9%) | 2.09 | 78 (95%) | 4 (5%) | - | - | 0.05 | 82 |
| GFP | 189 (100%) | - | - | - | 0.00 | 170 (90%) | 19 (10%) | - | - | 0.10 | 189 |

"Dataset" indicates experiment types with E. coli expressing NSs and/or GFP as indicated. For each dataset, data are collated from 3 separate cultures, each starting from a single colony. "Mean" indicates the mean number of MLOs per cell in each dataset. n is the number of cells analysed for each dataset.

with KL A disassemble at slightly lower temperature (39.9 °C) compared to those with KL B in vitro (40.9 °C, Supplementary Figs. 45–47). Instead, it may result from the greater degree of enzymatic degradation experienced by NS B (Supplementary Figs. 29, 30). A reduction in NS valency is also consistent with the broader melting transition seen in NS B compared to the A design. Different from A NSs, a small fraction (~20%) of cells expressing B MLOs do not re-assemble condensates when cooled, possibly due to nanostructure misfolding or failure to re-fold upon cooling. For both A and B NSs, condensate fluorescence intensity appears to increase after the heating and cooling ramps (Fig. 4A(i), B(i)), which may be attributed to thermally induced alterations in membrane permeability, facilitating greater intracellular accumulation of FLAP dyes. We further note that melting temperatures measured in vivo are substantially higher than those we observe in vitro, where the MLOs melted at 39.9–40.9 °C (Supplementary Fig. 45–47, 52). This deviation likely originates from intracellular molecular crowding promoting condensation[92] or difference in ionic conditions between the in vitro transcription buffer and the cytoplasm.

We then demonstrate temperature-triggered release of captured proteins by heating MLOs that had sequestered GFP. When cells expressing the A$_{AP3}$ NSs and GFP were briefly heated to 70 °C for 2 minutes, the GFP and nanostar fluorescent signal dispersed throughout the cell. Upon cooling, the MLOs reassembled at the poles, with GFP relocalising accordingly (Fig. 4C(i), (ii) and Supplementary Figs. 53, 54). Approximately 10% of the cells retained the ability to grow and divide following thermal treatment (Supplementary Fig. 55). Growth and division is found to resume after a 3-hour delay, which is likely the time required for bacterial recovery[93].

## Discussion

We demonstrate the expression of RNA nanostructures capable of forming non-natural membraneless organelles in live *E. coli*. Sequence selectivity enables the production of non-mixing and addressable organelle types with distinct composition, while embedding aptamers facilitates protein uptake. The genetically encoded, synthetic organelles are liquid-like and exhibit upper critical solution temperature behaviour, enabling reversible disassembly and reassembly upon heating and subsequent cooling. The system provides valuable insights on the biophysics of biomolecular condensation in prokaryotes, as we show by probing organelle number, spatial distribution, fluidity and response upon division. Compared to platforms reliant on engineered proteins and peptides, or simpler RNA repeats, the nanostructured motifs offer advantages in terms of rational and modular design. The observation that the RNA nanostars retain programmability and functionality within the complex intracellular environment is noteworthy, particularly for expression in *E. coli*, which, despite its biotechnological relevance, poses substantial challenges due to rapid RNA degradation. By overcoming the hurdle of RNA instability, our contribution paves the way to deploying nanostructured RNA condensates to metabolic engineering and biomanufacturing, leveraging the possibility of sequestering arbitrary proteins or nucleic acids within the condensates. One may indeed aim to optimise biocatalytic pathways by (co)localising enzymes and/or substrates, enhancing expression yields by sequestering toxic products or intermediates or improving separation and purification pipelines. Beyond industrial applications, the synthetic RNA condensates may be used to modulate biological functions in vivo by influencing gene regulation, buffering stress responses, and generally mimicking native phase-separated assemblies. For example, the condensates could be engineered to sequester transcriptional regulators or RNAs to alter gene expression profiles, protect key transcripts under stress, or serve as synthetic analogues of bacterial RNA granules[6]. These capabilities will add to the significant opportunities already offered by genetically-encoded RNA nanostructures, which have been deployed to regulate

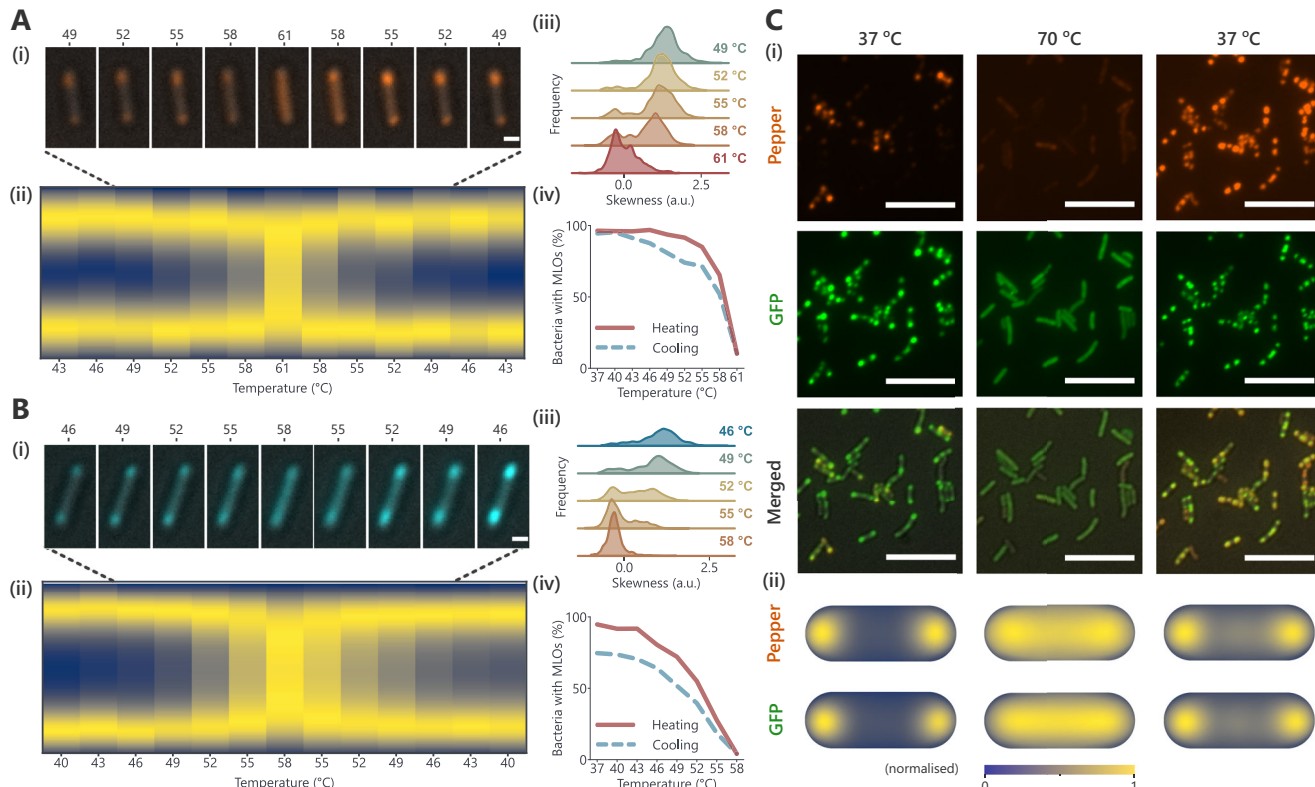

**Fig. 4 | Synthetic RNA MLOs reversibly disassemble and reassemble in *E. coli* upon temperature cycling. A** Reversible disassembly and reassembly of A MLOs. (i) Merged epifluorescence and bright-field micrographs showing disassembly and reassembly of A MLOs in a single cell. Scale bar, 1 μm. (ii) Population-averaged maps of Pepper (NS) fluorescence as a function of temperature upon heating and subsequent cooling. (iii) Cell-population distribution of the values of skewness computed for the fluorescence-intensity distributions of Pepper signal in individual *E. coli* on heating and cooling. (iv) Percentage of single cells containing MLOs as extracted from the skewness distributions (Methods). **B** Reversible disassembly

and reassembly of B MLOs. Data are presented as in **A**, using Broccoli fluorescence to image NSs. **C** Disassembly and reassembly of $A_{AP3}$ MLOs with captured GFP. (i) Confocal micrographs of samples at 37 °C (left), after heating to 70 °C (middle), and after cooling back to 37 °C (right). Top: Pepper (NS) fluorescence; middle: GFP fluorescence; bottom: merged Pepper fluorescence, GFP fluorescence, and bright-field. Scale bars, 10 μm. (ii) Population-averaged maps of Pepper (top) and GFP (bottom) fluorescence at the temperatures shown in (i). In **A**–**C**, fluorescence maps and quantitative analyses are based on *E. coli* from 3 independent cultures, each starting from a single colony. Source data are provided as a Source Data file.

transcription[94], translation[95], and metabolic pathways[96,97]. The platform could further be adapted to other specialised prokaryotic expression strains, including *Bacillus subtilis*, *Pseudomonas putida* and *Lactococcus lactis*. We expect that, taken together, these characteristics of robustness, modularity, structural programmability and interaction selectivity will establish nano-structured RNA condensates as a prominent tool for next-generation cellular engineering.

## Methods
### Chemicals and reagents
All reagents were obtained from Merck unless otherwise specified. *E. coli* BL21(DE3) and XL10-Gold cells were obtained from Agilent. DH5α cells were purchased from Thermo Fisher Scientific. TBI was purchased from Tocris Bioscience. HBC620 was purchased from MedChem Express. DFHBI-1T was purchased from Cayman Chemical. Recombinant GFP was purchased from Caltag Medsystems. Pico-Surf fluorinated oil was purchased from Sphere Fluidics. CellBrite Fix 640 membrane dye was purchased from Biotium. The T7-FlashScribe™ Transcription Kit was purchased from Cellscript. The RNeasy Mini kit and Bacteria Protect Reagent were obtained from Qiagen. GeneRuler 100 bp DNA Ladder and RiboRuler Low Range RNA Ladder were obtained from Thermo Fisher Scientific. LB-agar plates, glycerol, ultrapure water, and SOC media were obtained from the technical support team of the Department of Chemical Engineering and Biotechnology, University of Cambridge. Linear DNA strands were obtained from

Integrated DNA Technologies (IDT) as gBlocks. Geneframes were obtained from Thermo Fisher Scientific. Glass capillaries were purchased from VitroCom. Nunc MicroWell 96-Well Optical-Bottom Plates with Polymer Base (Cell culture, black, Surface Treatment: Cell Culture, Bottom: Flat) were obtained from Thermo Fisher Scientific. The pET plasmid (53137) was obtained from Addgene. The NEBuilder HiFi kit (E5520S) and the Monarch miniprep kit were obtained from New England Biolabs (NEB).

### Expression and characterisation of RNA nanostars in vitro
In vitro transcription reactions were prepared at room temperature using the T7-FlashScribe™ Transcription Kit and 40 nM of the relevant linear DNA template with T7 terminator or 5 nM of the relevant plasmid (for samples used in PAGE analysis). The final reaction mixture contained 9 mM of each ribonucleotide triphosphate, 0.05 units per μL RNase inhibitor, and 10 mM dithiothreitol. In addition, the reaction contained 50 μM of the Broccoli aptamer-binding dye DFHBI-1T, and 0.5 μM of the Pepper aptamer-binding dye HBC620. Mixtures were loaded into borosilicate glass capillaries with rectangular cross-section (0.2 mm × 2 mm, VitroCom) and capped with paraffin oil. Capillaries were then immediately placed at 37 °C and incubated for 4 hours, until imaging. Epifluorescence micrographs were obtained using a Nikon Eclipse Ti2-E inverted microscope equipped with a Hamamatsu ORCA-Flash4.0 V3 sCMOS camera, a tuneable light source (Lumencor SPEC-TRA X LED engine), and a Plan Apo λ 20 × 0.75 N.A. dry objective. The

HBC620 dye was excited using a 590 nm LED source with an LED-DA/FI/TR/Cy5-B filter, 10 ms excitation time, and fluorescence was recorded through an FF01-624/40 emission filter. DFHBI-1T was excited using a 470 nm LED source with an LED-DA/FI/TR/Cy5-B filter, 500 ms excitation time, and fluorescence was recorded through an EM510 emission filter. To determine the melting temperature of NS A and NS B, samples were incubated for 15 h at 30 °C and subsequently heated using a Peltier-controlled microscope stage (Temikra Ltd.). The temperature was increased from 30 °C to 50 °C and then cooled back to 30 °C. During this process, the temperature was adjusted by 1 °C every 5 min, with imaging performed after each temperature change. For directly comparing the fluorescence intensity of in vitro condensates with that recorded for *E. coli* (Supplementary Fig. 18), the in vitro transcription reaction was performed as described above, except for replacing DFHBI-1T with 10 µM TBI, as used for *E. coli*, followed by a 15 h incubation at 30 °C. For the in vitro transcription of NS A$_{AP3}$, the reaction was prepared as described above with the addition of 750 nM recombinant GFP. The transcription mixture was then encapsulated within water-in-oil droplets by mixing 22 µL of the solution with 90 µL of 2% w/w Pico-surf fluorinated oil and vortexing at 2,500 rpm for 30 s. The top layer, containing the emulsion droplets, was extracted, loaded into capillaries, and incubated at 37 °C for 8 h before being imaged on the Nikon Eclipse Ti2-E inverted microscope. The settings described above were used for imaging the Pepper channel (NS A$_{AP3}$), while GFP fluorescence was imaged using the 470 nm LED source with an LED-DA/FI/TR/Cy5-B filter and 2 ms excitation time. To induce thermal disassembly of the in vitro A$_{AP3}$+GFP samples, the loaded capillaries were heated using a Peltier microscope stage (Temikra Ltd.) to 20 °C, 37 °C and 40 °C, and subsequently cooled to 20 °C. To test the behaviour of NSs in buffer mimicking ionic conditions in *E. coli* (Supplementary Fig. 9), transcribed NSs were purified by Monarch Spin RNA Cleanup Kit (NEB), then 500 nM NS was added to 15 µL of buffer (40 mM Tris, pH 8.0, 200 mM KCl, 1 mM MgCl$_2$). The solution was incubated at 37 °C in 384-Well Optical-Bottom Microplate, black, non-treated PS surface (ThermoFisher) for 1 h before epifluorescence imaging.

**Image analysis of NS condensates in vitro.** Images (16-bit) were extracted from .nd2 (Nikon) or .czi (Zeiss) files. Fluorescence intensities reported in Supplementary Fig. 8 and for the comparison of in vitro and in vivo fluorescence intensities were quantified as the mean pixel intensity on mask-segmented images. Segmentation masks were generated using Otsu thresholding on flattened images, which were then applied to the corresponding raw images for intensity measurements. For melting-temperature analysis (Supplementary Fig. 45), raw 16-bit images were used to calculate the ratio between the standard deviation of the pixel values and the mean pixel intensity ($\sigma$/I). The melting and condensate-formation temperatures were determined as the points of intersection between the ratio curves and the plateau, defined as the mean of data points from 41 °C onwards, plus a constant of 0.05. To generate the displayed micrographs, images were rescaled to facilitate visualisation by dividing the pixel intensities by the intensity of the brightest pixels (99.9–100th percentile), and then exported as coloured micrographs. For micrographs showing condensate melting, condensate formation, and TBI dye comparisons, the brightest pixels across the entire image set were used for division. For all other micrographs, the brightest pixels within each individual image were used for division. The Python packages tifffile and numpy were used for handling and processing the microscopy images, scikit-image was used for image masking and thresholding, Pillow was used for micrograph export, and Matplotlib was used for creating plots.

## Plasmid construction

**Nanostar design.** RNA NSs A, Ā, B, and B̄ were adapted from Ref. 69. Each construct consists of a four-way junction with 25 bp double-stranded RNA arms. One arm in each design contains a FLAP, Pepper for A and Ā[74], and Broccoli for B and B̄[76]. Each arm in NS A ends with a KL with sequence 5'-AUCGCCAAA-3', while arms in NS B terminate with KL 5'-AGUCGACAA-3'. The KL sequences were designed in Ref. 69 to ensure similar binding strength while retaining mutual orthogonality and self-complementarity. NSs Ā and B̄ have KL nucleotide sequences scrambled to eliminate self-complementarity. NS A$_{AP3}$ additionally features the GFP-binding AP3 aptamer in one of the arms, opposite to the one hosting the Pepper aptamer[89]. NS arms that include fluorescent or GFP-binding aptamers retain KLs, ensuring that the constructs remain tetravalent. All NS designs were verified with Nupack[98] and Kinefold[99]. Sequences of all NS designs are reported in Supplementary Table 1.

**Plasmid design and construction.** To construct plasmids expressing single NSs (A, Ā, B, B̄, A$_{AP3}$), the natural T7 terminator sequence[80] was appended to the 3' end of the NS sequences. The NS+terminator sequences were cloned into a pET plasmid with an ampicillin resistance gene (Addgene #53137), using the NEBuilder HiFi kit, following the manufacturer's instructions. To construct plasmids expressing two NSs (A+B, A+B̄, Ā+B, or Ā+B̄), the natural T7 terminator was appended to the 3' end of NSs A and Ā, while a T7 terminator variant[80] was used for NSs B, and B̄. Different terminators were used to reduce the chances of non-specific interactions between the two constructs induced by the terminators. As for single NSs, the resulting sequences were cloned into the pET plasmid (Addgene #53137) using the NEBuilder HiFi kit. A 44 bp linker separates the two NS sequences on the plasmid. All plasmid maps are provided in Supplementary Figs. 10, 11. The GFP-expressing plasmid was constructed by cloning the eGFP sequence from pEGFP-N1 (Takara) into the pACYCDuet-1 backbone (Novagen), from which LacI had been removed. Both DNA fragments were obtained as linear sequences and assembled using the NEBuilder HiFi kit. The plasmids containing ampicillin resistance genes were propagated with XL-10 Gold cells, while plasmids containing chloramphenicol resistance genes were propagated in DH5$\alpha$ cells. Plasmids were extracted from the cloning strains using the Monarch miniprep kit. The sequences of all of the constructed plasmids were verified by whole-plasmid sequencing (Full Circle Labs).

## RNA nanostars expression in *E. coli*

Throughout all experiments, antibiotics matching the relevant resistance genes were used, namely ampicillin (50 µg mL$^{-1}$), and chloramphenicol (34 µg mL$^{-1}$).

**Transformation of BL21(DE3) strains.** To transform BL21(DE3) bacteria with plasmids containing the ampicillin resistance gene, 10 ng of the target plasmid were mixed with 10 µL of BL21(DE3) competent cells and incubated on ice for 15 min. The samples were plated on LB-agar containing carbenicillin and incubated at 37 °C for 16 h. To transform BL21(DE3) bacteria with plasmids containing the chloramphenicol resistance gene, or for co-transformation with plasmids containing different antibiotic resistance genes, 10 ng of each plasmid was added to 10 µL of BL21(DE3) competent cells and incubated on ice for 30 min. The samples were placed in a 42 °C water bath for 30 s, followed by 2 min incubation on ice, and addition of 250 µL Super Optimal broth with Catabolite repression (SOC) medium. The samples were placed in a shaking incubator (Multitron Standard, Infors HT, 37 °C, 225 rpm) for 1 h, before plating on LB-agar with matching antibiotics, and incubating at 37 °C for 16 h. LB-agar plates with transformed cells were stored at 4 °C for up to two weeks before RNA expression experiments.

**RNA and protein expression in *E. coli*.** A single colony was transferred to 5 mL M9 minimal medium, supplemented with 0.4% glycerol, 0.2% casamino acids, 2 mM MgCl$_2$, 100 µM CaCl$_2$ (hereby referred to as M9 medium with supplements), and matching antibiotics. The culture was

grown for 4 h (225 rpm, 37 °C) in a shaking incubator, to reach an $OD_{600}$ around 0.6–0.8, measured by spectrophotometer (SmartSpec Plus, Bio-Rad). IPTG (0.4 mM), TBI (10 µM), HBC620 (5 µM) were added to 2 mL of the sample, and the samples were further incubated for 1 h (225 rpm, 37 °C) in a shaking incubator. TBI dye was used instead of DFHBI-1T for imaging in cells due to its improved photostability in vivo, which makes it better suited for cellular imaging applications[76]. 10 µL of the sample was diluted with 10 µL of the M9 medium with supplements, including the same concentrations of dyes, inducer, and antibiotics. The samples were stained with 0.2 µL of CellBrite Fix 640 membrane dye (1000 ×) at 25 °C for 10 minutes, before transferring to agarose pads for imaging.

**Extraction of RNA from *E. coli*.** Bacterial samples were prepared as described above. RNA was extracted from bacteria using the Qiagen RNeasy Mini kit with Bacteria Protect Reagent, following the manufacturer's instructions. In the cell lysis step, the cells were sonicated for three cycles at maximum speed for 30 s, with 1 min rest between the cycles (Soniprep 150, MSE supplies LLC).

**Quantifying NS expression in *E. coli* with RT-qPCR.** Extracted RNA samples were reverse-transcribed and amplified using the Takyon No ROX SYBR (Eurogentec) mastermix and Takyon One Step converter (Eurogentec) in MicroAmp Optical 96-well reaction plates with optical adhesive covers (Applied Biosystems) on the QuantStudio 6 Pro qPCR (ThermoFisher). 60 ng of total RNA per sample was loaded into each well in a total reaction volume of 7.1 µL. Three biological replicates and three technical replicates were used for each sample. The cycle number at which a sample's fluorescence signal crosses a predefined threshold (Cq) was recorded for each sample. To quantify the relative amount of target transcripts, a housekeeping gene (*adk*) was used as a reference. The relative transcript number (FD) is then calculated by:

$$\Delta Cq = Cq_{Gene} - Cq_{Reference} \tag{1}$$

$$FD = 2^{-\Delta Cq} \tag{2}$$

Primers B_fwd and B_rev targeted NS B, while primers adk_fwd and adk_rev targeted *adk*. Primers were selected to amplify the first half of the NS, to facilitate the detection of cleaved constructs (Supplementary Fig. 30). The primer sequences were: B_fwd: 5′-GCACAGTGCTATGAGTGTC-3′, B_rev: 5′-CGACTGTGGCATACAGCGAC-3′, adk_fwd: 5′-ATTCTGCTTGGCGCTCCGGG-3′, adk_rev: 5′-CCGTCAACTTTCGCGTATTT-3′. The thermocycling parameters were as follows: 48 °C (10 min), followed by 95 °C (3 min), then 40 cycles of 95 °C (10 s) and 60 °C (1 min). The temperature ramp rate was 1.6 °C s⁻¹ for all steps. To construct the calibration curve, in vitro transcribed NS RNA was purified using the Monarch Spin RNA Cleanup Kit (NEB). The concentration of purified NS was quantified by UV absorbance using a Nanodrop One spectrophotometer. Calibration curves were created by measuring the Cq values at different NS concentrations. The intracellular concentration of NS was estimated by dividing the amount of extracted RNA by the overall intracellular volume. The overall intracellular volume calibration curve created using known purified RNA concentrations was used to calculate the concentration of RNA per mL of culture or per cell. To compute the latter, we estimated the total number of cells in the culture using optical density (OD) (1 OD = (1.0 ± 0.3) × 10⁸ colony forming units mL⁻¹, based on 4 independent measurements), and multiplied by the expected single-cell volume of 3 µm³, obtained by approximating the cell as a cuboid with dimensions 1 µm × 1 µm × 3 µm.

**Preparation of agarose pads for imaging.** For *E. coli* imaging on agarose pads, we prepared 2% molten low-gelling-temperature agarose in M9 media with supplements and added IPTG (0.4 mM), TBI (10 µM), HBC620 (5 µM), and matching antibiotics. The molten agarose solution was kept at 50 °C, and 100 µL was added to a smaller gene frame (10 mm × 10 mm) adhered to a microscope slide (Clear ground 0.8–1.0 mm, Fisher Scientific). The molten agarose was sandwiched between two glass slides and left at 25 °C for 10 min to solidify. The solidified agarose was transferred to larger gene frames (17 mm × 28 mm) and adhered to a glass slide. 2 µL of the prepared cells was added to each agarose pad, and after 5 minutes at 25 °C, a coverslip (24 mm × 60 mm, No.1, Epredia) was adhered to the other side of the gene frame.

**Confocal imaging of *E. coli***
The prepared agarose pads were imaged with a Zeiss LSM 800 confocal microscope, using a Plan-Apochromat 63 × 1.4 N.A. Oil DIC M27 objective. The fluorescent signal from the Pepper aptamer was acquired using a 561 nm laser set to 1.0% intensity and a Master Gain of 500 V, with an emission window of 580 to 650 nm. The fluorescent signal from the Broccoli aptamer and GFP was acquired using a 488 nm laser set at 2.0% intensity for A+B̄ and 1.0% for all other samples and a Master Gain of 500 V, with an emission window of 470 to 580 nm for all samples other than for A+B̄, which had a restricted emission window of 470 to 530 nm. The settings for samples A+B̄ were optimised to minimise bleed-through of the Pepper fluorescence in the Broccoli channel. While this fluorescence is weak and generally not noticeable in other samples, it becomes noticeable here because A is concentrated at MLOs, and B̄ is mostly evenly distributed (Supplementary Fig. 56). The image frame size was set to 1192 pixels × 1192 pixels with 16-bit depth. Line-averaging was enabled and set to 4 ×. The scan mode along the *x* direction was selected to be bidirectional. The scanning rate was set to produce a scanning time of 1.77 µs per pixel. This protocol was also used for imaging the in vitro samples when comparing fluorescence intensities between the in vitro and *E. coli* samples.

**FRAP of *E. coli*.** The 561 nm laser was set at 100% intensity for photobleaching A MLOs, while the 488 nm laser was set at 100% intensity for photobleaching B MLOs and GFP-containing MLOs. A total of 25 A MLOs, 12 B MLOs, and 25 GFP-containing MLOs were chosen for photobleaching, with the entire condensate bleached in each case. A total of 15 A MLOs, 8 B MLOs, 15 GFP-containing MLOs were chosen as non-photobleached controls. Fluorescence recovery was monitored at 10 s intervals. Condensate intensity was quantified as the mean intensity of the 10 brightest pixels within the MLO.

**Epifluorescence imaging of *E. coli***
**Time-lapse images of *E. coli*.** Time-lapse images of cells expressing NS A were acquired with a Ti2 Eclipse Epifluorescent microscope (Nikon), equipped with an sCMOS camera (Hamamatsu ORCA-Flash4.0 V3), a tunable light source (Lumencor SPECTRA X LED engine), and CFI Plan Apo λD 60 × 1.42 N.A. oil immersion objective. The Pepper aptamer was excited using a 555 nm LED with an LED-DA/FI/TR/Cy5-B filter, and fluorescence was collected through an FF01-624/40 emission filter with a 500 ms exposure time. Images were taken every 5 minutes. Samples were incubated at 37 °C using a heating stage (Temikra Ltd.).

**Temperature-dependent dissolution and re-assembly of MLOs.** To study the melting and reforming of MLOs in bacteria, bacteria with NS A or NS B were prepared for imaging as described above, except that 2% agarose low EEO was used for the agarose pads. *E. coli* samples on agarose pads were attached to a Peltier microscope stage (Temikra Ltd.) equipped with a sapphire microscope window to ensure optical transparency and good thermal conduction (UQGOptics), and imaged with a Ti2 Eclipse Epifluorescent microscope (Nikon), equipped with an sCMOS camera (Hamamatsu ORCA-Flash4.0 V3), a tuneable light source (Lumencor SPECTRA X LED engine), and a CFI Plan Apo λD 60 × 1.42 N.A. oil immersion objective.

The Pepper aptamer was excited using a 555 nm LED with an LED-DA/FI/TR/Cy5-B filter, and fluorescence was collected through an FF01-624/40 emission filter with a 500 ms exposure time. The Broccoli aptamer was excited using a 470 nm LED with an LED-DA/FI/TR/Cy5-B filter, and fluorescence was collected through an EM510 emission filter with a 2 s exposure time. The samples were heated from 37 °C to either 58 °C or 61 °C with increments of 3 °C every 2 min. Samples were then cooled from their maximum temperature to 37 °C with increments of 3 °C every 2 min. For bacteria co-expressed with NS$_{AP3}$ and GFP, cells were imaged using the same Pepper aptamer settings as for Nanostar A, while the GFP signal was captured on the 470 nm LED with an LED-DA/FI/TR/Cy5-B filter, and fluorescence was collected through an EM510 emission filter with a 50 ms exposure time. Cells were heated to 37 °C, 70 °C and cooled to 37 °C, with 2 min incubation periods between each temperature.

### Analysis of *E. coli* micrographs

Images and metadata were extracted from .czi (Zeiss) or .nd2 (Nikon) files using pylibCZIrw and Nd2Reader, respectively. The brightfield image underwent flat-field correction and non-local means denoising. Initial binary masks of the cells were generated using a calculated threshold and adjusted manually if needed. Small holes were filled, and objects with an area smaller than half of a typical *E. coli* cell (2 μm × 0.5 μm) were removed. Since the algorithm struggled to segment individual bacteria within microcolonies, bacteria inside these regions were filtered out based on object proximity. Objects on image borders were discarded. The remaining objects were filtered based on eccentricity (higher than 0.85), aspect ratio (between 2.5 and 4.5), area (smaller than 7.07 μm$^2$), and width of *E. coli* (smaller than 1.5 μm)[100–103]. The final segmented set was further filtered manually via an interactive GUI. For $R^2$ calculation, pixel intensities from one fluorescent channel were compared against those from another, yielding the coefficient of determination ($R^2$). To compute kurtosis and skewness for each bacterium, pixel intensities in segmented single cells were normalised so that the top 5% of pixels had a value of 1. These pixels were then removed, and kurtosis and skewness of the pixel-intensity distribution were calculated using Scipy[104]. To determine the number of MLOs per bacterium, each segmented bacterium was rotated so that its long axis aligned horizontally. To minimise interpolation loss, pixels were subdivided into 16 smaller pixels before rotation. Intensities were then averaged vertically to obtain a linear profile, from which peaks were detected using Scipy. The sensitivity of peak detection was manually adjusted based on noise levels. For heatmaps of fluorescence intensity, horizontally aligned bacteria were scaled to a standardised size of 48 pixels × 130 pixels, and pixel intensities were averaged across aligned positions. For Fig. 2B(ii), cells were aligned with their most fluorescent region oriented upwards. This was determined by the Pepper channel for cells co-expressing nanostar A and B/B̄, and by the Broccoli channel for cells co-expressing nanostar Ā and B. To determine locations of MLOs, the condensates were segmented using adaptive thresholding (cv2)[105]. MLO positions and sizes were validated by comparing them to peaks detected at similar locations within the cell to prevent misclassification. The MLO position was recorded as the highest-intensity pixel within the mask. For quantification of condensate and dispersed-phase intensities, local intensity maxima were identified in each segmented cell. Only cells in which the detected number of peaks matched the expected condensate count were retained. Around each peak, a 2D Gaussian was fitted and an elliptical condensate mask was generated, which defined the condensate region. The integrated condensate intensity, $I_{cond}$, was computed as the sum of pixel intensities within this region. The dispersed-phase intensity, $i_{disp}$, was calculated as the mean intensity in the remaining cell area after excluding the condensate part. $i_{tot}$ was computed as the mean intensity within the entire cell area. All analyses were performed using Python 3.10, and visualisations were generated with Matplotlib[106] and Seaborn[107]. To generate the displayed fluorescence micrographs, images were rescaled to facilitate visualisation by dividing the pixel intensities by the intensity of the brightest pixels (99.9-100th percentile). To generate the displayed bright-field micrographs, intensities were rescaled by subtracting the intensity of the dimmest pixel and dividing by the range between the brightest and dimmest pixels. For micrographs showing condensate melting and condensate formation, the brightest pixels across the entire image set were used for division. For all other micrographs, the brightest pixels within each individual image were used for division.

### Analysis of the temperature-dependent dissolution and reassembly of MLOs

Micrographs were analyzed fully automatically to obtain segmented bacteria. Heatmap progression across different temperatures was determined by extracting the maximum intensity along the long axis of standard heatmaps, computed as discussed above. The skewness of the intensity distribution of the pixel intensity within single cells was used to monitor the progression of dissolution and reassembly. To estimate the proportion of cells with and without MLOs, the distribution of skewness values across the cell population at a given temperature was fitted using a mixture of two Gaussians (using skimage), with one Gaussian fitting the sub-population of cells with high skewness (featuring condensates) and the second fitting the sub-population of cells with low skewness (lacking condensates)[108]. For each cell, the probability of belonging to either sub-population was calculated. If the probability of belonging to the MLO-forming distribution exceeded 0.5, the cell was classified as MLO-forming. To estimate $T_m$ for single cells, cells with skewness above 0.25 at 37 °C (indicating the presence of condensates) were selected, and their skewness-temperature profiles were fitted with a sigmoid curve. The inflection point of the sigmoid fit was taken as the estimated $T_m$.

### Growth and RNA production curves in bulk cultures

To characterise growth and RNA production in bulk cell samples, *E. coli* from single colonies were grown in M9 media with supplements for 4 h as described above. *E. coli* samples were then diluted to OD$_{600}$= 0.05. 100 μL of the diluted solutions were loaded in a Nunc MicroWell 96-Well Optical-Bottom well plate, together with TBI (10 μM), HBC620 (5 μM), and matching antibiotics. The samples were cultured at 37 °C in a plate reader (Tecan M2000). Every 7 minutes, the plate was shaken (orbital) for 5 s with 3.5 mm amplitude, followed by absorbance and fluorescence measurements. For absorbance, wavelength was set to 600 nm and bandwidth was set to 9 nm. For detecting Broccoli fluorescence, we set the excitation wavelength to 470 nm, the excitation bandwidth to 9 nm, the emission wavelength to 520 nm and the emission bandwidth to 20 nm. For Pepper fluorescence, we set the excitation wavelength to 570 nm, excitation bandwidth to 9 nm, emission wavelength to 620 nm and emission bandwidth to 20 nm. When the OD$_{600}$ reached 0.7, we added 0.4 mM IPTG to the samples requiring induction, before continuing the experiment with unchanged settings.

### RNA sequencing of *E. coli*

*E. coli* expressing A or B nanostars were induced for 1 h as described above. Cells were pelleted and shipped to Full Circle Labs for RNA extraction, poly(A) tailing, and nanopore sequencing. To identify start and end positions of RNA raw reads[109], the reads were aligned to the expected RNA sequence with the Needleman-Wunsch algorithm[110], and reads with alignment scores corresponding to ≥7 matched nucleotides were retained for further analysis. Due to known imprecision in base-calling near the 5′ ends of direct nanopore RNA reads[111], fragment start positions were defined as the first aligned nucleotide in a stretch of at least 3 consecutive matches between the read and the expected sequence.

## Polyacrylamide gel electrophoresis

The gels were prepared with a 19:1 w/v acrylamide:bis-acrylamide solution in Tris-Borate-EDTA (TBE) buffer. Acrylamide fractions (w/w) of 6% and 8% were used for native and denaturing gels, respectively. The denaturing gel contained 7 M urea. For both native and denaturing PAGE, samples transcribed in vitro were diluted 1:10, while for *E. coli* RNA extracts, 500 ng of RNA were used. Samples were then mixed with 0.83 μL of the DNA Gel Loading Dye (6 ×) for native PAGE and 2.5 μL of the RNA Gel Loading Dye (2 ×) for denaturing PAGE before loading a total of 5 μL of mixture into the respective lanes. For the denaturing gels, samples were heated to 70 °C for 10 min and subsequently cooled on ice for 3 min before loading. For native gels, one lane contained 1.25 μL of the GeneRuler 100 bp DNA Ladder, while 1.25 μL of the RiboRuler Low Range RNA Ladder was used for denaturing gels. For both native and denaturing gels, sample stacking was performed by running the gel at 40 V for 20 min immediately after loading. The native gels were then run at 150 V for 60 min, while the denaturing gels were run at 270 V for 60 min. All gels were run in TBE. Gels were stained after electrophoresis with 1 × SYBR Gold nucleic acid gel stain and imaged using a Syngene G:BOX Chemi XRQ gel documentation system. Lane intensities were quantified using ImageJ[112]. The full, uncropped gel images are shown in Supplementary Fig. 57.

## Statistics & reproducibility

No statistical method was used to predetermine sample size. No data were excluded from the analyses, except cells in large clumps that cannot be clearly segmented. The experiments were not randomized. The Investigators were not blinded to allocation during experiments and outcome assessment.

## Reporting summary

Further information on research design is available in the Nature Portfolio Reporting Summary linked to this article.

## Data availability

Supplementary Figs., supplementary tables, supplementary notes, and gene sequences are available in the supplementary information. Source data is provided with this paper. The raw sequencing data generated in this study have been deposited in the NCBI Bioproject database under accession code PRJNA1307508. Microscopy data in support of this publication are available in the Apollo Repository under ID: ADE9D896-7339-4018-9013-004F682B9EA1 [https://doi.org/10.17863/CAM.123436]. Source data are provided with this paper.

## Code availability

The data analysis code used in this publication is available at https://github.com/adamkni/Condensate_Segmentation[113].

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

## Acknowledgements

The team thanks Yuval Elani, Elisa Franco and Hirohide Saito for useful discussions and the support staff of the Department of Chemical Engineering and Biotechnology, University of Cambridge, especially Danita Pearson, for their dedicated assistance. The work was supported by the Horizon 2020 research and innovation programme (ERC-STG No 851667-NANOCELL, L.D.M., L.M., C.F.), Royal Society University Research Fellowship (UF160152, URF\R\221009, L.D.M), UK Biotechnology and Biological Science Research Council (BBSRC) through a Japan Interdisciplinary Partnership Award Plus (BB/X012557/1) and from JST ASPIRE (JPMJAP24B4) (L.D.M., B.N. and M.T.), MEXT/JSPS KAKENHI (Nos. JP24H00070 and JP25H01361, M.T.), Biotechnology and Biological Sciences Research Council through a BBSRC Discovery Fellowship (BB/X010228/1, R.R.S.), Ernest Oppenheimer Fund studentship (School of Physical Sciences, University of Cambridge, M.D.), The University of Cambridge Harding Distinguished Postgraduate Scholars Programme and the UK Engineering and Physical Sciences Research Council

(EPSRC, 2928387, A.K.), the Department of Chemistry at Imperial College London (G.F), EPSRC (EP/S023518/1, S.P.N).

## Author contributions

B.N., C.F., M.D., and A.K. contributed to the design, execution, and analysis of experiments. L.M. contributed to the pipeline for image segmentation and analysis. G.F. and S.P.N. contributed to nanostar design for *E. coli* expression. R.R.-S., G.C., M.T., and P.C. provided technical support, contributed to experimental methodology, and assisted with manuscript preparation. L.D.M. conceived and supervised the study, coordinated collaborations, and finalized the manuscript. All authors discussed the results and contributed to the final version of the paper.

## Competing interests

B.N., C.F, M.D., A.K. and L.D.M., through the University of Cambridge, have filed a patent application in the UK Intellectual Property Office, which includes disclosure of all inventions described in this manuscript, provisional application serial number 2510955.4, filed on 7 July 2025. The other authors declare no conflict of interest.
