## [Peer Review File · Nature Communications]

Expression of nano-engineered RNA organelles in bacteria

Corresponding Author: Professor Lorenzo Di Michele

Version 0:

Reviewer comments:

Reviewer #1

(Remarks to the Author)

The manuscript entitled “Expression of Nano-engineered RNA Organelles in Bacteria” presents a highly interesting study that leverages RNA nanotechnology to construct programmable, orthogonal, membrane-less organelles in *E. coli*. These organelles assemble co-transcriptionally, recruit proteins via aptamers, and can reversibly dissolve and reassemble through thermal cycling. Notably, this work represents the first demonstration of nanostar RNA condensates expressed in a living system, as well as specific protein recruitment to a designed RNA organelle. While the study is impressive and conceptually novel, several issues require attention before publication.

1. For the stability of NS transcripts in *E. coli*, the half-life of NS transcripts is unknown. Circularization is a powerful method to enhance RNA transcript stability. Have you tried this approach to improve stability? What is the local concentration of NS in *E. coli*? Is it comparable to the level in the *in vitro* transcriptional NS condensate? Does the NS condensate exhibit a similar Broccoli aptamer intensity when the same TBI concentration is applied?
2. In Extended Figs. 8 and 9, nanopore sequencing was used to confirm NS transcription in *E. coli*. However, Extended Figs. 8A and 9A show relatively low and gradually increasing fragment coverage for Arm 1a of both NS A and NS B. Moreover, there is no expected cleavage site at Arm 1a (Extended Figs. 8B and 9B). The author should provide a clearer explanation for this observation.
3. In the section “Expression of orthogonal membrane-less organelles in *E. coli*,” Fig. 3 should be referenced as Fig. 2. In the section “Selective protein recruitment in RNA membrane-less organelles,” Fig. 4 should be referenced as Fig. 3.
4. A magnesium and sodium titration experiment confirmed that NS condensate formation requires concentrations exceeding 20 mM Mg²⁺ or 20 mM Na⁺ (Nat. Commun., 2024, 15, 6244). However, the free ion concentrations in *E. coli* could be much lower. The author needs to explain why this condensate can form without disassembly.
5. Regarding the explanation: “In addition, fluorescence bleed-through from the Pepper (NS A) to the Broccoli (NS B) channels, observed *in vitro* (Extended Data Fig. 1), may contribute to the appearance of B clusters, despite optimizing imaging settings (Methods and Supplementary Fig. 21).” This explanation is inconsistent. When NS A and NS B condensates are co-expressed, these two-color channels work well, so why would applying the same settings now cause crosstalk between them?
6. For “By fitting the skewness distributions, we can extract a temperature dependent fraction of cells hosting MLOs, which shows a small hysteresis between heating and cooling ramps, likely caused by slow assembly kinetics or delays in thermal equilibration in the samples”, However, this explanation may be more applicable to non-living systems. In living cells, protein crowding is present, and these proteins could denature after a heating-cooling cycle, altering the local environment of the MLOs.
7. “Importantly, the cells remained viable and were able to grow and divide after heating and cooling.” This conclusion appears to be in stark contrast with previously reported findings (Journal of Applied Microbiology Symposium Supplement 2000, 88, 798-898; FEMS Microbiol Lett 2006, 260, 100-105.)
8. A recent publication on such co-transcriptional condensates in crowded environments should be added to the background in the introduction (<https://www.nature.com/articles/s41467-025-63445-8>).
9. As a side note, please avoid the ultracompact and narrow formatting of the PDF to simplify reviewing.

finally, congratulations to the authors, I enjoyed the manuscript very much.

(Remarks on code availability)

Reviewer #2

(Remarks to the Author)

The manuscript "Expression of nano-engineered RNA organelles in bacteria" by Ng et al. presents a novel and compelling study on the development of methods to express unnatural condensates in *E. coli* cells using RNA nanostructures. The authors provide a thorough characterization of condensate morphologies derived from various nucleic acid designs, demonstrating that specific RNA-RNA base-pairing interactions mediate condensation. They also perform experiments on the co-expression of two distinct types of droplet compartments within the same bacterial cell, the selective sequestration of specific proteins within these droplets, and the thermo-responsive behavior of the condensates that mirrors observations from *in vitro* analogs.

To my knowledge, this work represents the first demonstration of nucleic acid nanotechnology-driven condensates *in vivo*, opening exciting avenues for nanoscale design to modulate biological function as well as for probing the intricate cellular environment. The results are generally strong and well-supported. However, enhancing the characterization, analysis, and discussion could significantly improve the manuscript's clarity, rigor, and potential impact. These suggestions are listed roughly in order of priority:

1. Condensate Dynamics and Equilibration Mechanisms

- The dynamics of condensate formation during RNA expression remain unclear. A time-resolved analysis of condensation kinetics would elucidate whether condensation approaches near-equilibrium morphologies or is maintained in a far-from-equilibrium state.
- After cell division, it is noted that droplet sizes are often unequal (one droplet being approximately halved). Does the size difference change over time? Does the smaller droplet dissolve like one would expect for an Ostwald ripening mechanism or are droplets actively maintained at the poles?

2. Quantitative Characterization Using Thermodynamically Relevant Metrics

- The authors should clarify why droplet volume fraction is not used as a quantitative measure. In my view, a key strength of nanostar condensates is that they are theoretically very well-described. Estimating condensate volume fraction from measured droplet sizes relative to cell volume and comparing this to expression yields (i.e. bulk average nanostar concentration) would allow for a more complete thermodynamic interpretation.
- In Extended Figure 7, the data show an average absorbance and fluorescence as a function of time. Could this be used to calibrate the expression yield using fluorescence? More precisely, a gel-based assay at multiple time points to quantify "complete" nanostar expression would be even better.

3. Impact of the Cellular Interior

- The physical characteristics of the nucleoid and its energetic landscape should be discussed in relation to condensate formation. What is the energetic cost of creating a void within the nucleoid? How might the nucleoid's properties influence the transport and equilibration of nanostar species across the cell?

4. Biological Implications and Functional Impact

- While the introduction highlights the design potential of nucleic acid nanotechnology, the manuscript would benefit from a brief but explicit discussion of potential biological functions that nanostar condensates could modulate *in vivo* in the conclusion.
- Additionally, a brief comparison with other nucleic acid nanotechnologies and their *in vivo* applications would improve the potential impact.

5. Co-expressed Orthogonal Condensates

- Figure 2Bii's microscopy images are hard to interpret, especially in areas where cells are tightly packed, and they do not seem to reflect the diagram that shows one of each condensate at both poles. Additionally, could the authors clarify whether chromatic aberration was corrected and what calibration methods were used?
- Extended Figure 1B suggests possible mixing of condensate types. Is there direct evidence of mixing or interaction between droplet species *in vivo*?
- Extended Figure 10 shows droplets predominantly localized to one side of the cell when coexpressed. What might explain this asymmetric distribution? Could slight condensate mixing provide a nucleation point for B onto A?

6. Temperature Dependence of Condensates

- In Figure 4Ai, the uniform RNA control appears brighter than condensate-containing cells after normalization. Since normalization by maximum intensity varies across cells, this may be misleading; raw intensity comparisons would be more informative.
- What is the *in vitro* melting temperature of the condensates? Does melting temperature correlate with cellular expression level or fluorescence intensity? Given theoretical predictions, higher nanostar concentration should increase the melting temperature until the critical point, after which it declines. Exploring this relationship experimentally would enhance the thermodynamic analysis.

(Remarks on code availability)

README is blank.

Version 1:

Reviewer comments:

Reviewer #1

(Remarks to the Author)

Great effort during the revisions. This is a great study and is ready for publication.

(Remarks on code availability)

Reviewer #2

(Remarks to the Author)

I appreciate the authors' detailed and thorough response, which included substantial new experimental work, data analysis, and writing that significantly enhance the quality of the manuscript. My concerns have been addressed, and I support publication.

(Remarks on code availability)

We thank the Editor and the Reviewers for taking the time to consider our submission and for their positive and constructive feedback to improve it. Below, we address point-by-point the questions raised, and we enclose updated versions of the manuscript and SI, where modifications have been made accordingly and are highlighted in **red**.

In addition to discussing the points suggested by the Reviewers and providing additional data (figures and tables), we have fixed various typos and standardised the method through which fluorescence images are normalised (simple linear rescaling), to facilitate cross-experiment comparisons. We further replaced the uncropped gel images in Supplementary Fig. 48, as we noted that, in our initial submission, we had accidentally included images of gels acquired during the optimisation stages, rather than the final gels produced in optimised conditions and used in the main text. We apologise for this mistake, which has now been fixed.

As requested, a sentence mentioning source data has been included in the data availability section, and the source data used to generate the figures are enclosed as an Excel spreadsheet. In addition, we also included a link to a permanent, freely accessible data repository containing raw data. The link will become active before the manuscript is published in its final form.

In this document, points raised by the Reviewers are highlighted in *blue*, while our answers are presented in black. In “...” we presented quoted excerpts of the text highlighting in **red** the modified portions of the revised manuscript. All modified and newly included graphical elements, quoted in this response, are included for quick reference. References quoted in the response are listed at the bottom of the document, and their numbering does not match that used in the main text or SI.

Reviewer 1

The manuscript entitled “Expression of Nano-engineered RNA Organelles in Bacteria” presents a highly interesting study that leverages RNA nanotechnology to construct programmable, orthogonal, membrane-less organelles in E.coli. These organelles assemble co-transcriptionally, recruit proteins via aptamers, and can reversibly dissolve and reassemble through thermal cycling. Notably, this work represents the first demonstration of nanostar RNA condensates expressed in a living system, as well as specific protein recruitment to a designed RNA organelle. While the study is impressive and conceptually novel, several issues require attention before publication.

We thank the Reviewer for their positive assessment on our work, and for recommending publication following the corrections addressed below.

1. For the stability of NS transcripts in E. coli, the half-life of NS transcripts is unknown. Circularisation is a powerful method to enhance RNA transcript stability. Have you tried this approach to improve stability?

Because we observe robust condensate formation in the vast majority of cells, we have not explored circularisation as a strategy to enhance RNA transcript stability.

Nonetheless, we fully agree that circularisation could be a valuable approach, and will consider it in subsequent studies, should greater stability be required. We mentioned this possibility in the revised manuscript.

“In addition, stability may be enhanced by circularising the constructs to prevent digestion by exonucleases [1].”

What is the local concentration of NS in E. coli? Is it comparable to the level in the in vitro transcriptional NS condensate?

We thank the Reviewer for raising this interesting point. In the revised manuscript, we used RT-qPCR to estimate the concentration of NS B expressed in *E. coli* both before induction and at 1 h post-induction, the timepoint at which cells were imaged. These results are presented in the new Supplementary Fig. 14, included below, where we report: A) NS concentrations relative to the mRNA of a housekeeping gene; B) Absolute NS concentration in the cell culture; and C) An estimate of the average intra-cellular NS concentration, obtained by using optical density (OD) to determine cell counts ($1 \text{ OD} = (1.0 \pm 0.3) \times 10^8$ colony forming units mL^{-1} , based on 4 independent measurements). At 1 h post-induction, RT-qPCR estimates the intracellular NS concentration to be $16 \pm 5 \mu\text{M}$ (7 biological replicates). We can use the measured intracellular concentration to roughly estimate the average volume of the condensed phase. Under the simplified assumptions that, within the condensates, each NS occupies the volume equal to that of a sphere with radius equal to the NS arm length ($\sim 8 \text{ nm}$, see [2]), the volume per NS is approximately $2.11 \times 10^{-6} \mu\text{m}^3$. Approximating the cell as a cuboid with dimensions $1 \mu\text{m} \times 1 \mu\text{m} \times 3 \mu\text{m}$, a concentration of $16 \mu\text{M}$ corresponds to about 31,000 intracellular NSs. Assuming that most expressed NSs are gathered in condensates, the total condensate volume is roughly $6.7 \times 10^{-2} \mu\text{m}^3$. If this volume is equally divided between two spherical condensates per cell, this back-of-the-envelope calculation yields an expected condensed diameter of $\sim 400 \text{ nm}$. These length scales are not inconsistent with microscopy observations, with the caveat that an accurate determination of condensate size is hindered by diffraction-limited resolution. Intracellular NS concentration is also similar to values obtained with IVT expression, namely $13 \pm 4 \mu\text{M}$ for NS B, measured from three independent IVT reactions after 3 hours of incubation (Supplementary Fig. 14D).

We have now included the new information in the main text:

“RT-qPCR was used to quantify NS expression levels, yielding an intra-cellular concentration of $16 \pm 5 \mu\text{M}$ for NS B 1 hour post-induction (Supplementary Figs. 13, 14, Methods). By assuming that most NSs accumulate in the condensates, and approximating the per-NS volume in the condensed phase as that of a sphere with radius equal to the NS arm length ($\sim 8 \text{ nm}$ [2]), we can roughly estimate the total condensate volume from the NS concentration. By further assuming that this volume is equally divided between two condensates, we estimate a mean condensate diameter

of ~ 400 nm, compatible with microscopy observations. Intra-cellular NS concentration is comparable with in vitro expression yields, measured as 13 ± 4 μM 3 hours from starting the reaction (Supplementary Fig. 14).”

RT-qPCR was used here as it is the most widely adopted technique for measuring intracellular RNA concentrations due to its sensitivity, specificity, and relative ease of use. This said, RT-qPCR is not without limitations, as outcomes can be influenced by variability in RNA extraction, secondary structure, enzymatic processing (Extended Data Figs. 8A-B, 9A-B) and post-transcriptional modifications. These considerations are included in the caption of Supplementary Fig. 14.

Full details on the RNA extraction and RT-qPCR protocols are included in the revised Methods section. We also include new Supplementary Fig. 13, reporting calibration data used for absolute RNA quantification. We have commented on these results in the revised main text.

Supplementary Figure 13: RT-qPCR Nanostar concentration calibration curve. RT-qPCR calibration curve showing the relationship between C_q values and the natural logarithm of RNA concentration for NS B. Linear fitting produced $C_q = -1.59 \times \ln(\text{NS concentration}/\text{ng } \mu\text{L}^{-1}) + 11.8$. See Methods for experimental details.

Supplementary Figure 14: Concentration of NSs expressed in *E. coli* as analysed by RT-qPCR. (A) Bar graphs of the relative concentration of NS B, normalised to the housekeeping gene *adk*. (B) Absolute concentrations of NS B per mL of cell culture. (C) Intracellular concentrations of NS B, estimated by dividing the data in (B) by the overall intracellular volume. To compute the latter, we estimated the total number of cells in the culture using optical density (OD) ($1 \text{ OD} = (1.0 \pm 0.3) \times 10^8 \text{ colony forming units mL}^{-1}$, based on 4 independent measurements), and multiplied by the expected single-cell volume of $3 \mu\text{m}^3$, obtained by approximating the cell as a cuboid with dimensions $1 \mu\text{m} \times 1 \mu\text{m} \times 3 \mu\text{m}$ (see Methods). The height of the bars are the mean values of three technical replicates (RT-qPCR measurements) performed on each biological replicate grown from independent colonies. For the 1 h induction condition, seven biological replicates were analysed; for the 0 h induction condition, three biological replicates were analyzed. Error bars indicate standard error of the mean. (D) Side-by-side comparison of the intracellular concentration of NS B and that of NS B expressed in vitro 3 h after starting the reaction. For RT-qPCR, a calibration curve produced using known concentrations of purified RNA was used to calculate the concentrations in (B) and (C) (Supplementary Fig. 13). NS concentration in the in vitro sample was determined with UV absorbance using a Nanodrop ONE spectrophotometer, following RNA purification. We note that, while RT-qPCR is widely adopted for measuring intracellular RNA concentrations due to its sensitivity, specificity, and relative ease of use, outcomes can still be influenced by variability in RNA extraction, secondary structure, and enzymatic processing (Extended Data Figs. 8A-B, 9A-B) and post-transcriptional modifications.

Does the NS condensate exhibit a similar Broccoli aptamer intensity when the same TBI concentration is applied?

We thank the Reviewer for this question. To address it, we carried out additional IVT experiments where NS B condensates were stained with 10 μ M TBI, the same concentration used for the *E. coli* experiments. The confocal images were acquired following the same imaging protocol used for *E. coli* experiments and subjected to the same post-processing steps (Methods). Image segmentation reveals that the mean pixel intensity of in vitro NS B condensates is just 23% higher than that of the condensates expressed in *E. coli*. This suggests comparable fluorescence intensity in the Broccoli channel between the in vitro and in vivo samples that, in turn, hints at similarities in FLAP response and overall NS concentration in the condensed phase. For additional clarity, micrographs showing side-by-side comparison of in vitro and in vivo expressed NS B condensates are included in new Supplementary Fig. 16.

We comment on these results in the revised main text:

“When identical staining and imaging settings are applied, typical fluorescence intensity recorded from condensates expressed in vivo closely matches the one measured in vitro, suggesting similar FLAP response and NS concentration in the condensed phase (Supplementary Fig. 16).”

Supplementary Figure 16: Confocal micrographs of in vitro and in vivo NS B condensates stained using the same TBI concentration (10 μ M). The micrographs were obtained using the same imaging protocol and subjected to the same post-processing settings (Methods). The segmented images exhibit comparable fluorescence, with the mean pixel intensity of NS B condensates expressed (i) in vitro being 23% higher than that of NS B condensates expressed (ii) in *E. coli*. This suggests similarities in FLAP response and overall NS concentration in the condensed phase. Scale bars, 20 μ m.

2. In Extended Figs. 8 and 9, nanopore sequencing was used to confirm NS transcription in E. coli. However, Extended Figs. 8A and 9A show relatively low and gradually increasing fragment coverage for Arm 1a of both NS A and NS B.

The relatively low and gradually increasing coverage for Arm 1a—corresponding to the 5' end of the nanostar transcript—is likely attributed to known limitations in basecalling accuracy near the 5' end of direct RNA nanopore reads [3]. The RNA is sequenced from 3' to 5', and the helicase enzyme that controls translocation through the pore is at a distance equivalent to 10-15 nt from the pore itself. Thus, when the enzyme releases the last nucleotide at the 5' end, the final 10-15 nucleotides of the strand are rapidly driven through the pore, which prevents their accurate reading. To clarify the observed coverage pattern, we have now included a mention of this effect in the relevant figure captions.

“Relatively low coverage at the 5' end (Arm 1a) is consistent with known basecalling limitations in direct RNA nanopore reads. Specifically, the helicase enzyme controlling translocation through the nanopore sits at a distance equivalent to 10-15 nt from the sensing region, causing the final 10-15 nt of a strand being sequenced to rapidly translocate once the 5' is released by the enzyme, preventing accurate reading [3].”

Moreover, there is no expected cleavage site at Arm 1a (Extended Figs. 8B and 9B). The author should provide a clearer explanation for this observation.

The absence of cleavage at Arm 1 is indeed notable. While predictive tools for RNase cleavage sites are currently lacking, one possible explanation is that the T7 terminator, which protrudes from the stem structure, may sterically hinder RNase access and protect this region from degradation. Similarly, cleavage on Arm 2 is not observed, which could be due to the steric hindrance from the FLAP on Arm 2. We have now included a discussion of this in the manuscript.

“In both NS designs, cleavage was not detected on NS arms that include the FLAP or the T7 terminator, possibly due to these structural features hindering RNase access [4]. Both NSs, however, feature a cleavage site in the terminator loop itself, which is thus likely to be detached in many transcripts (Extended Data Figs. 8A-B, 9A-B). The apparent protective effect of altering the duplex structure of NS arms could be leveraged to design against degradation.”

3. In the section “Expression of orthogonal membrane-less organelles in E. coli,” Fig. 3 should be referenced as Fig. 2. In the section “Selective protein recruitment in RNA membrane-less organelles,” Fig. 4 should be referenced as Fig. 3.

The typo has now been corrected.

4. A magnesium and sodium titration experiment confirmed that NS condensate formation requires concentrations exceeding 20 mM Mg^{2+} or 20 mM Na^+ (*Nat. Commun.*, 2024, 15, 6244). However, the free ion concentrations in *E. coli* could be much lower. The author needs to explain why this condensate can form without disassembly.

In *E. coli*, the total intracellular unbound magnesium concentration is estimated to be around 1 mM [5], while potassium levels range from 200 to 400 mM [6]. To test for the stability of the condensates under relevant conditions, we performed control experiments in which we purified in vitro-expressed NSs and redissolved them in a buffer with 1 mM magnesium chloride and 200 mM potassium chloride. As shown in the new Supplementary Figure 8, condensate formation was readily observed. This suggests that the concentration of magnesium and potassium found in *E. coli* can support condensate growth. Additional cellular factors, such as molecular crowding, may further promote condensate assembly.

We have included the new data and accompanying discussion in the manuscript main text:

“Encouragingly for in vivo expression, robust condensation was observed in buffers mimicking the intracellular ionic conditions encountered in *E. coli* (200 mM K^+ and 1 mM Mg^{2+} [5, 6]) (Supplementary Fig. 8).”

Supplementary Figure 8: Epifluorescence micrographs of in vitro expressed NS incubated in buffer mimicking the intracellular ionic environment of *E. coli*. Epifluorescence micrographs of (i) NS A. (ii) NS B, after 1 hour incubation in buffer containing 1 mM MgCl₂ and 200 mM KCl. Scale bars, 200 μm.

5. *Regarding the explanation: “In addition, fluorescence bleed-through from the Pepper (NS A) to the Broccoli (NS \bar{B}) channels, observed in vitro (Extended Data Fig. 1), may contribute to the appearance of \bar{B} clusters, despite optimizing imaging settings (Methods and Supplementary Fig. 21).” This explanation is inconsistent. When NS A and NS B condensates are co-expressed, these two-colour channels work well, so why would applying the same settings now cause crosstalk between them?*

We thank the Reviewer for the valuable suggestion that fluorescence bleed-through from the Pepper (NS A) to the Broccoli channel (NS \bar{B}) might not be the most plausible explanation for the observed NS \bar{B} clusters in the A+ \bar{B} samples.

To address this issue, we applied image segmentation to micrographs of in vitro-expressed condensates and reported the data in new Supplementary Fig. 7, shown below for quick reference. First, we quantified fluorescence emission in the Broccoli channel (normally used to image NSs B and \bar{B}) in samples containing only the condensate-forming NS A, labelled with Pepper FLAPs (Supplementary Fig. 7A). The amount of bleed-through from the Pepper channel into the Broccoli channel was minimal.

We then performed the same analysis on samples expressing A NSs together with dispersed \bar{B} NSs and, unsurprisingly, we observed substantially higher signal in the Broccoli channel (Supplementary Fig. 7B). As noted in our initial submission, the signal within A condensates was higher than in the surrounding background, an effect initially ascribed to bleed-through.

In Supplementary Fig. 7C, we then computed the difference, δ , between the Broccoli signal measured within A condensates and that measured in the surrounding background. Comparing A-only and A+ \bar{B} samples, δ is substantially higher in the latter. Two effects could contribute to this: (i) bleed-through from the Pepper dye being stronger within A-type condensates than in the surrounding solution, and (ii) a difference in \bar{B} NS concentration between the A-type condensates and the background. If (i) were the only effect—i.e. if the concentration of \bar{B} within A condensates were equal to that in the background—then similar δ values would be observed in both A and A+ \bar{B} samples. Instead, the much greater δ in A+ \bar{B} suggests that \bar{B} accumulates within A condensates, causing the increased Broccoli signal. We surmise that this accumulation arises from weak affinity between the scrambled KLS on \bar{B} and the A KLS.

Such non-specific affinity is not observed in condensate-forming B NSs. In A+B samples, Broccoli fluorescence is higher in the background than in A condensates, consistent with excluded volume effects that lower the concentration of the dispersed B phase within A condensates relative to the surrounding solution. The absence of A-B condensate adhesion further supports the lack of non-specific A-B affinity (Supplementary Fig. 7B).

In the revised manuscript, we have commented on these new observations where relevant, and referred to the new Supplementary Figure.

“The greater abundance of non-specific clusters seen when non-sticky NS are co-expressed with sticky motifs, particularly for $A+\bar{B}$ expression, may be due to MLO-templated clustering of the otherwise soluble NSs. The soluble NSs may indeed weakly partition within, or coat the specifically-assembled MLO, or the condensates may serve as heterogeneous nucleation sites for non-specific aggregation. Either process may be facilitated by weak affinity between the soluble and sticky NSs due to base pairing, misfolding, enzymatic degradation, or the action of native RNA-binding proteins. **This interpretation is supported by the evidence of weak partitioning of soluble NS \bar{B} in NS A condensates observed in vitro (Extended Data Fig. 1), and quantified in Supplementary Fig. 7 (Methods).**”

We have included a summary of the discussion provided here in the caption of the new Supplementary Fig. 7.

Supplementary Figure 7: Quantification of the Broccoli signal observed in A-type condensates. (A) Mean Broccoli fluorescence within and outside (bulk phase) NS A condensates in NS A-only samples. The Broccoli signal in the denser NS A condensates differs only slightly from that in the bulk phase, indicating that fluorescence bleed-through from the Pepper channel into the Broccoli channel is minimal. (B) Mean Broccoli fluorescence within and outside NS A condensates in $A+\bar{B}$ and $A+B$ samples. In $A+\bar{B}$ samples, Broccoli fluorescence is higher within NS A condensates than in the bulk, whereas in $A+B$ samples, fluorescence is higher in the bulk phase. The latter observation is consistent with excluded volume effects that lower the concentration of NS B within A condensates relative to the bulk phase. (C) Difference, δ , in mean Broccoli fluorescence intensity between NS A condensates and the surrounding bulk phase in A-only and $A+\bar{B}$ samples. The larger δ in $A+\bar{B}$ samples suggests accumulation of NS \bar{B} within A condensates, likely due to weak affinity between the scrambled KFs of NS \bar{B} and the KFs of NS A.

6. For “By fitting the skewness distributions, we can extract a temperature dependent fraction of cells hosting MLOs, which shows a small hysteresis between heating and cooling ramps, likely caused by slow assembly kinetics or delays in thermal equilibration in the samples”, However, this explanation may be more applicable to non-living systems. In living cells, protein crowding is present, and these proteins could denature after a heating-cooling cycle, altering the local environment of the MLOs.

We agree with the insightful comment by the Reviewer, and modified the manuscript to include this argument.

“By fitting the skewness distributions, we can extract a temperature-dependent fraction of cells hosting MLOs, which shows a small hysteresis between heating and cooling ramps. This may be caused by slow assembly kinetics, delays in thermal equilibration, or changes in protein crowding due to protein denaturation during heating, which could alter the local environment of the MLOs (Fig. 4A(iv) and Supplementary Fig. 40).”

7. “Importantly, the cells remained viable and were able to grow and divide after heating and cooling.” This conclusion appears to be in stark contrast with previously reported findings (*Journal of Applied Microbiology Symposium Supplement 2000, 88, 798-898; FEMS Microbiol Lett 2006, 260, 100-105.*)

We thank the Reviewer for the comment, which prompted us to re-analyse our data on post-thermal-treatment cell viability. Overall, the cell population resumes growth, as we had previously reported. However, we have now quantified that only a small fraction of cells, approximately 10%, retain the ability to grow and divide and is responsible for population growth. Those surviving cells required roughly 3 hours before division was observed, indicating that stress from heat exposure significantly slowed down division.

In Ref. [7], quoted by the Reviewer, the cell lethality was quantified using a live/dead stain to show compromised cell membranes as well as DAPI to show compromised nucleoids. Neither are the most accurate measure of cell death, as cells with compromised membrane [8] and nucleoid [9] can still recover. Bacterial cells can be in a metabolically viable but not culturable state, that over time can be recovered [10], consistent with our observation of delayed division post heating.

Ref. [11], also quoted by the Reviewer, analyses the effect of heating on bacteria in conditions quite different from ours. This includes bacteria in homogenised meats as well as in skimmed milk, where they note a 6-decimal reduction of cells after heating at 70 °C for 2 mins. They note that there might be differences to heat resistance when cells are in a different media, if a different strain is used or if a different heating treatment is used, which may be the case for our experiments. Additionally this paper does not investigate further the long term recovery of these cells, but does

reference work from Ref. [12], reporting on the recovery of bacteria when heated at high temperatures for a short period. Ref. [12] notes that some bacteria can repair sublethal damage after 4+ hours, reflecting our results post heating.

In summary, we observe that a fraction of the cells are able to survive the heating treatment and later resume division, leading to the observed population growth.

We have revised the main text to reflect our quantification of the percentage of viable cells, and discuss our observation in the context of literature.

“Approximately 10% of the cells retained the ability to grow and divide following thermal treatment (Supplementary Fig. 46). Growth and division is found to resume after a 3-hour delay, which is likely the time required for bacterial recovery [10].”

8. A recent publication on such co-transcriptional condensates in crowded environments should be added to the background in the introduction (<https://www.nature.com/articles/s41467-025-63445-8>).

We have expanded the introduction to include this relevant literature:

“To overcome this roadblock, here we deploy RNA nanostructures, which retain much of the programmability of DNA constructs [2, 13–15], but can be synthesised in vivo [16] and in crowded environments that mimic cellular conditions [17].”

9. As a side note, please avoid the ultracompact and narrow formatting of the PDF to simplify reviewing.

Thank you for the feedback. For our initial submission, we used the Springer Nature LaTeX template, which enforces this formatting. In the marked main text prepared for this resubmission, we edited the template to enforce double spacing, which we hope facilitates reviewing. The unmodified, single-spacing template has been retained for the unmarked version of the main text to facilitate manuscript production.

finally, congratulations to the authors, I enjoyed the manuscript very much.

Thank you. We appreciate your help in further improving the manuscript.

Reviewer 2

The manuscript “Expression of nano-engineered RNA organelles in bacteria” by Ng et al. presents a novel and compelling study on the development of methods to express unnatural condensates in E. coli cells using RNA nanostructures. The authors provide a thorough characterisation of condensate morphologies derived from various nucleic acid designs, demonstrating that specific RNA-RNA base-pairing interactions mediate condensation. They also perform experiments on the co-expression of two distinct types of droplet compartments within the same bacterial cell, the selective sequestration of specific proteins within these droplets, and the thermo-responsive behaviour of the condensates that mirrors observations from in vitro analogs. To my knowledge, this work represents the first demonstration of nucleic acid nanotechnology-driven condensates in vivo, opening exciting avenues for nanoscale design to modulate biological function as well as for probing the intricate cellular environment. The results are generally strong and well-supported. However, enhancing the characterisation, analysis, and discussion could significantly improve the manuscript’s clarity, rigor, and potential impact. These suggestions are listed roughly in order of priority:

We thank the Reviewer for the positive assessment of our manuscript and for considering it to be written to a high standard.

1. Condensate Dynamics and Equilibration Mechanisms. The dynamics of condensate formation during RNA expression remain unclear. A time-resolved analysis of condensation kinetics would elucidate whether condensation approaches near-equilibrium morphologies or is maintained in a far-from-equilibrium state. After cell division, it is noted that droplet sizes are often unequal (one droplet being approximately halved). Does the size difference change over time? Does the smaller droplet dissolve like one would expect for an Ostwald ripening mechanism or are droplets actively maintained at the poles?

Quantitative estimation of condensate size and (even more so) morphology are challenging due to condensate dimensions being close to or below the diffraction-limited resolution of our microscope. Yet, under the assumption that the density of NSs, and therefore of the fluorescent dye, is uniform within the condensates, the fluorescence signal integrated across a condensate, I_{cond} , is expected to be proportional to the condensate volume.

Following to this strategy, we estimated the time-dependent ratio between the integrated fluorescence intensity of the condensates present at the two cell poles (smaller over larger condensate, $r = I_{\text{cond}}^{\text{small}}/I_{\text{cond}}^{\text{large}}$), and used it as a proxy for the volume ratio. We tracked r for a few cells until they approached the division stage, as reported in the new Supplementary Fig. 18, included below for quick reference. In all cases, r increases over time, starting from ~ 0.3 – 0.6 and approaching ~ 0.6 – 0.9 . Evidence that condensate asymmetry originating from cell division is progressively eliminated suggests that Ostwald ripening is not prominent. Rather, the smaller condensate formed at the new cell pole tends to grow over time, with its size approaching

that of the larger, older condensate. Studies of similar DNA nanostar systems in vitro also exclude Ostwald ripening as a dominant mechanism for condensate growth [18].

We have included a comment on this observation in the revised main text:

“In some instances, the central condensates then appear to split between the two daughter cells, which emerge with two polar condensates of asymmetric intensity (Fig. 1D(ii) and Extended Data Fig. 4(i)). Over time, smaller condensates grow towards parity with the larger ones, progressively reducing size asymmetry (Supplementary Fig. 18). This evidence excludes Ostwald ripening as a dominant mechanism for condensate coarsening, consistent with observations made on DNA nanostar condensates in vitro [18].”

Supplementary Figure 18: Ratio of condensate sizes in single cells expressing NS A over time. The ratio r was calculated as $r = I_{\text{cond}}^{\text{small}} / I_{\text{cond}}^{\text{large}}$, where $I_{\text{cond}}^{\text{small}}$ is the integrated fluorescence intensity of the smaller MLO and $I_{\text{cond}}^{\text{large}}$ is the integrated fluorescence intensity of the larger MLO. Each line represents one cell, tracked until the onset of cell division. The area of MLOs was selected manually using ImageJ. Images were acquired every 5 minutes.

2. Quantitative Characterisation Using Thermodynamically Relevant Metrics. The authors should clarify why droplet volume fraction is not used as a quantitative measure. In my view, a key strength of nanostar condensates is that they are theoretically very well-described. Estimating condensate volume fraction from measured droplet sizes relative to cell volume and comparing this to expression yields (i.e. bulk average nanostar concentration) would allow for a more complete thermodynamic interpretation.

We thank the Reviewer for the insightful comment. As discussed above, we do not feel we can reliably estimate condensate size, due to the droplets being often close to or below the diffraction limit. We can, however, use the integrated fluorescence intensity within a condensate, I_{cond} , as a proxy for its size, as done in point 1. In addition we can, with some accuracy, compute the average fluorescence intensity, i_{disp} , in the centre of the cells away from the condensates, which is proportional to the concentration of NSs in the dispersed (gas) phase. In the new Supplementary Fig. 15A we show a scatter plot of i_{disp} vs I_{cond} , and observe that the former is nearly independent on the latter for both A and B-type condensates. This finding is consistent with the system being at or near thermodynamic equilibrium, which would imply the concentration of dispersed NSs coexisting with the condensates being independent of condensate size. Expectedly, the mean fluorescence intensity measured across the entire cell, i_{tot} , proportional to the overall NS concentration, is strongly linearly correlated with I_{cond} , as shown in Supplementary Figure 15B.

In the revised main text we have commented on these observations, referring to the new Supplementary Figure.

The main text now reads:

“Through image segmentation applied to single cells, we determined the integrated fluorescence intensity of individual condensates, I_{cond} , expected to be proportional to their volume. Similarly, we computed the average intensity in the condensate-free mid cell region, i_{disp} , proportional to the concentration of the dispersed NS phase, and the average intensity across the entire cell, i_{tot} , proportional to the overall NS concentration (Methods). In Supplementary Fig. 15, we show that i_{disp} is weakly correlated with I_{cond} ($R^2 = 0.21$ and 0.24 for A and B, respectively), consistent with the notion that the NS behave similarly to an equilibrium phase-separating system, in which the concentration of the dispersed phase is independent on the volume of the dense phase. In addition, consistently with the lever rule, i_{tot} is strongly linearly correlated with I_{cond} ($R^2 = 0.70$ and 0.83 for A and B, respectively).”

Supplementary Figure 15: Probing of the thermodynamic behaviour of NSs within cells. (A) Scatter plot of the average fluorescence intensity measured outside condensates (i_{disp}) against the integrated fluorescence inside segmented condensates (I_{cond}). i_{disp} is proportional to the concentration of dispersed NSs, while I_{cond} is expected to be proportional to condensate volume, under the assumption of uniform NS density and brightness in the condensed phase. Linear fits (dashed gray line) are shown with the following parameters: NS A (top) ($y = 6.4 \times 10^{-5}x + 4.3 \times 10^{-3}$, $R^2 = 0.21$) and NS B (bottom) ($y = 5.1 \times 10^{-5}x + 1.3 \times 10^{-2}$, $R^2 = 0.24$). Weak correlation between i_{disp} and I_{cond} is consistent with NS concentration in the dispersed phase being independent on condensate volume, as expected at thermodynamic equilibrium. (B) Scatter of the average fluorescence of the whole cell, i_{tot} , proportional to the overall NS concentration, against I_{cond} . Linear fits (dashed gray line) are shown with the following parameters: NS A (top) ($y = 2.0 \times 10^{-4}x + 4.9 \times 10^{-3}$, $R^2 = 0.70$) and NS B (bottom) ($y = 2.1 \times 10^{-4}x + 1.3 \times 10^{-2}$, $R^2 = 0.83$). At thermodynamic equilibrium, a linear dependence of condensate volume on overall NS concentration is consistent with the lever rule.

In Extended Figure 7 (platerreader), the data show an average absorbance and fluorescence as a function of time. Could this be used to calibrate the expression yield using fluorescence (total concentration of nanostars)? More precisely, a gel-based assay at multiple time points to quantify “complete” nanostar expression would be even better.

We agree with the Reviewer on the importance of quantifying NS expression levels. RT-qPCR was selected as the analysis method as it is the most widely adopted technique for measuring intracellular RNA concentrations due to its sensitivity, specificity, and relative ease of use.

We considered plate reader measurements less reliable, as one would need to perform calibrations with in vitro expressed (or reconstituted) NSs, whose fluorescence intensity (per NS) may differ from the in vivo case due to differences in dye availability or micro-environment.

The RT-qPCR results are presented in the new Supplementary Fig. 14. For *E. coli* expressing NS B, we report: A) NS concentrations relative to the mRNA of a housekeeping gene; B) Absolute NS concentration in the cell culture; and C) An estimate of the average intra-cellular NS concentration, calculated using optical density (OD) to determine cell counts ($1 \text{ OD} = (1.0 \pm 0.3) \times 10^8 \text{ colony forming units mL}^{-1}$, based on 4 independent measurements).

At 1 h post-induction, RT-qPCR yields an estimated intracellular NS concentration to be $16 \pm 5 \text{ }\mu\text{M}$ (7 biological replicates), similar to the value measured in vitro ($13 \pm 4 \text{ }\mu\text{M}$ 3 h from starting the reaction).

In the revised manuscript, we describe the new results in the main text:

“RT-qPCR was used to quantify NS expression levels, yielding an intra-cellular concentration of $16 \pm 5 \text{ }\mu\text{M}$ for NS B 1 hour post-induction (Supplementary Figs. 13, 14, Methods). By assuming that most NSs accumulate in the condensates, and approximating the per-NS volume in the condensed phase as that of a sphere with radius equal to the NS arm length ($\sim 8 \text{ nm}$ [2]), we can roughly estimate the total condensate volume from the NS concentration. By further assuming that this volume is equally divided between two condensates, we estimate a mean condensate diameter of $\sim 400 \text{ nm}$, compatible with microscopy observations (Fig. 1B(i)-C(i)). Intra-cellular NS concentration is comparable with in vitro expression yields, measured as $13 \pm 4 \text{ }\mu\text{M}$ 3 hours from starting the reaction (Supplementary Fig. 14).”

Supplementary Figure 13: RT-qPCR Nanostar concentration calibration curve. RT-qPCR calibration curve showing the relationship between Cq values and the natural logarithm of RNA concentration for NS B. Linear fitting produced $Cq = -1.59 \times \ln(\text{NS concentration}/\text{ng } \mu\text{L}^{-1}) + 11.8$. See Methods for experimental details.

Supplementary Figure 14: Concentration of NSs expressed in *E. coli* as analysed by RT-qPCR. (A) Bar graphs of the relative concentration of NS B, normalised to the housekeeping gene *adk*. (B) Absolute concentrations of NS B per mL of cell culture. (C) Intracellular concentrations of NS B, estimated by dividing the data in (B) by the overall intracellular volume. To compute the latter, we estimated the total number of cells in the culture using optical density (OD) ($1 \text{ OD} = (1.0 \pm 0.3) \times 10^8 \text{ colony forming units mL}^{-1}$, based on 4 independent measurements), and multiplied by the expected single-cell volume of $3 \mu\text{m}^3$, obtained by approximating the cell as a cuboid with dimensions $1 \mu\text{m} \times 1 \mu\text{m} \times 3 \mu\text{m}$ (see Methods). The height of the bars are the mean values of three technical replicates (RT-qPCR measurements) performed on each biological replicate grown from independent colonies. For the 1 h induction condition, seven biological replicates were analysed; for the 0 h induction condition, three biological replicates were analyzed. Error bars indicate standard error of the mean. (D) Side-by-side comparison of the intracellular concentration of NS B and that of NS B expressed in vitro 3 h after starting the reaction. For RT-qPCR, a calibration curve produced using known concentrations of purified RNA was used to calculate the concentrations in (B) and (C) (Supplementary Fig. 13). NS concentration in the in vitro sample was determined with UV absorbance using a Nanodrop ONE spectrophotometer, following RNA purification. We note that, while RT-qPCR is widely adopted for measuring intracellular RNA concentrations due to its sensitivity, specificity, and relative ease of use, outcomes can still be influenced by variability in RNA extraction, secondary structure, and enzymatic processing (Extended Data Figs. 8A-B, 9A-B) and post-transcriptional modifications.

3. *Impact of the Cellular Interior. The physical characteristics of the nucleoid and its energetic landscape should be discussed in relation to condensate formation. What is the energetic cost of creating a void within the nucleoid? How might the nucleoid's properties influence the transport and equilibration of nanostar species across the cell?*

We thank the Reviewer for the insightful question. In the revised submission we evaluated the energy cost of forming a condensate in the dense nucleoid region. This may provide a driving force for the migration of nascent condensates from the cells' centre towards the poles, as observed in Extended Data Fig. 3A, and ultimately help explain the overwhelming tendency of condensates to accumulate at the poles. The associated discussion is included in the new Supplementary Note 1:

Supplementary Note 1: Energetics of condensate formation within the nucleoid

The bacterial nucleoid is a dense, dynamic, and viscoelastic polymeric mesh composed primarily of DNA and associated proteins [19]. It occupies the central region of the cell due to ongoing replication and cohesion of sister chromosomes at the terminus region, which spatially anchors the nucleoid mid-cell [20]. Forming a condensate in the central region of the cell would create a void within the nucleoid, and thus have an associated energetic cost. To estimate this energy, we start from the work of Pelletier et al. [21]. In cells, the chromosome is compressed by the osmotic pressure exerted by cytosolic crowding agents that are excluded from the nucleoid. To imitate the effect of intracellular crowding, Pelletier et al. compressed the *E. coli* nucleoid in a microfluidic channel using a bead driven by optical tweezers, and showed the nucleoid's mechanical response can be modelled as an entropic spring

$$\frac{f}{A} = \left(\frac{R}{R_0}\right) - \left(\frac{R}{R_0}\right)^{-2},$$

where f is the compression force, $A = -2.04$ pN is a scaling constant, $R_0 = 10.1$ μm is the equilibrium length of the nucleoid in the microfluidic channel of width $w = 1.6$ μm , and R is the compressed nucleoid length. Here, lengths refer to the spatial extent of the nucleoid along the channel axis.

The compression pressure is thus given by

$$P = \frac{f}{w^2} = \frac{A}{w^2} \left[\left(\frac{R}{R_0}\right) - \left(\frac{R}{R_0}\right)^{-2} \right].$$

The energy required to create a void of volume ΔV , given an initial nucleoid volume V , can be estimated as

$$\Delta E = \int_V^{V+\Delta V} P dV = \int_V^{V+\Delta V} \frac{A}{w^2} \left[\left(\frac{R}{R_0}\right) - \left(\frac{R}{R_0}\right)^{-2} \right] dV.$$

For a small ΔV , one can assume that the pressure P is unaffected by the volume change, which allows us to simply express the energy cost of void formation as $\Delta E \approx P \Delta V$.

Pelletier et al. report that that a force of 100 pN is required to compress the nucleoid to its in vivo size ($R/R_0 \approx 0.1$), resulting in $P \approx 80 \text{ J m}^{-3}$ [21]. Using this value, we can estimate the energy cost of forming a spherical void of 50 nm in diameter as $\approx 4.2 \times 10^{-20}$ J, or $\approx 130 k_B T$. This energy cost is roughly 500-fold higher than the hybridisation free energy of the KL domain (8.8×10^{-23} J, estimated using Nupack [22]). This rough estimate suggests that the formation of nanostar clusters within the nucleoid region is strongly disfavoured, consistent with the observed, overwhelming tendency for pole accumulation.

Beyond energetic constraints, it is also likely that the nucleoid presents a significant steric barrier for transport of nanostars and nanostar clusters across the cell, as previously reported for protein aggregates [23]. Combined with the energy cost of deforming the nucleoid, these steric effects would make it unlikely for nanostar clusters formed in nucleoid-free regions to diffuse into the nucleoid.

Finally, we note that cell crowding can be modulated, for instance by using sub-lethal concentrations of antibiotics [24]. Changing cell crowding should impact the mechanical characteristics of the nucleoid, which may in turn influence condensate localisation.

In the main text, we comment on these results and refer to the new Supplementary Note 1:

“Pole localisation is likely to result from steric hindrance by the nucleoid located in the central section of the cell [25]. To test this hypothesis, in Supplementary Note 1, we use simple polymer physics arguments to estimate the energy cost of forming a small condensate (50 nm in diameter) within the nucleoid [21]. We find this penalty to be as high as $\sim 130 k_B T$, equivalent to $\sim 500\times$ the KL hybridisation free energy, which justifies the exclusion of the condensates from the central segment of the cells. In addition to the effect of the nucleoid, anisotropic confinement has also been hypothesised as contributing to pole accumulation of aggregates [26].”

“We propose that the rapid migration of mid-cell nascent condensates toward the poles is driven by the energetic cost associated with nucleoid deformation (Supplementary Note 1).”

4. Biological Implications and Functional Impact. While the introduction highlights the design potential of nucleic acid nanotechnology, the manuscript would benefit from a brief but explicit discussion of potential biological functions that nanostar condensates could modulate in vivo in the conclusion.

We agree with the Reviewer and have expanded our conclusion to include discussion of potential biological functions that nanostar condensates could modulate:

“Beyond industrial applications, the synthetic RNA condensates may be used to modulate biological functions *in vivo* by influencing gene regulation, buffering stress responses, and generally mimicking native phase-separated assemblies. For example, the condensates could be engineered to sequester transcriptional regulators or RNAs to alter gene expression profiles, protect key transcripts under stress, or serve as synthetic analogues of bacterial RNA granules [27].”

*Additionally, a brief comparison with other nucleic acid nanotechnologies and their *in vivo* applications would improve the potential impact.*

We thank the Reviewer for the suggestion. In the revised main text, we have expanded the conclusion to briefly discuss key functions demonstrating genetically encoded, and *in-vivo* expressed RNA nanostructures (scaffolds, RNA origami), which are the closest to the class of nucleic acid nanodevices of our condensate-forming nanostars:

“These capabilities will add to the significant opportunities already offered by genetically-encoded RNA nanostructures, which have been deployed to regulate transcription [28], translation [29], and metabolic pathways [30, 31].”

5. Co-expressed Orthogonal Condensates. Figure 2Bii’s microscopy images are hard to interpret, especially in areas where cells are tightly packed, and they do not seem to reflect the diagram that shows one of each condensate at both poles.

Thank you for the feedback. We agree with the comments and have made several improvements in the revised Fig. 2B, included below.

1. Microscope images with fewer microcolonies have been selected to improve clarity, and included in Fig. 2B (i).

2. The schematic illustration in Fig. 2B(ii) and 2D has been revised so that the number of condensates depicted now matches the measured average number of condensates per cell (Fig. 2B (iii)).

3. The average fluorescence intensity distribution heatmap (Fig. 2B (ii)) was originally constructed by using cells in random orientation along the longitudinal axis. As a result, although most cells contained a single condensate at one pole, the random orientation caused condensates to appear at both poles in the average image - half at the top and half at the bottom- creating an impression of bipolar condensate formation. We have now updated the protocol to align cells with their most fluorescent pole oriented upwards, which led to a more accurate representation of the condensate localisation pattern. The new analysis protocol has been outlined

in the revised methods section:

“For Fig. 2B(ii), cells were aligned with their most fluorescent pole oriented upwards. This was determined by the Pepper channel for cells co-expressing nanostar A and B/ \bar{B} , and by the Broccoli channel for cells co-expressing nanostar \bar{A} and B.”

For completeness, the heatmaps used for the initial submission, are retained as Supplementary Fig. 24, and mentioned in the main text:

“Similar to single-NS systems, condensates are preferentially located towards the poles, as evident from both average fluorescence maps (Fig. 2B(ii), Supplementary Fig. 24) and the locations of individual MLOs (Fig. 2C).”

Figure 2. Co-expression of orthogonal, non-mixing RNA MLOs in *E. coli*. (A) Schematic of the co-expression of A and B NSs in *E. coli*. A and B motifs have different kissing loop sequences and cannot interact by base-pairing, therefore forming separate MLOs in *E. coli*. Both NS sequences are cloned into a single plasmid for T7 RNA polymerase transcription, each controlled by its own promoter and a distinct terminator [32]. (B) Co-expression of NS pairs in *E. coli*. Top to bottom: A+B, A+B̄, Ā+B and Ā+B̄ NSs. (i) Confocal micrographs. Left: Pepper fluorescence (NSs A and Ā). Middle: Broccoli fluorescence (NSs B and B̄). Right: Merged Pepper fluorescence, Broccoli fluorescence, and bright-field. Scale bars: 10μm. (ii) Schematic representation (left) and population-averaged maps of Pepper fluorescence distribution (middle), and Broccoli fluorescence distribution (right) in *E. coli* (Methods). (iii) Percentage of *E. coli* containing 0, 1, 2 or 3 A or B MLOs per cell. (C) Scatter plot of the locations of MLOs in *E. coli* co-expressing both A and B NSs. Each symbol represents the position of a detected MLO (Methods). (D) Box plots of the values of kurtosis extracted from the pixel-value distribution of fluorescence intensity in single cells. Analysis is performed on (i) Pepper and (ii) Broccoli fluorescence for cells expressing A+B, Ā+B, A+B̄ and Ā+B̄ NSs (left to right). Outcomes of statistical tests are provided in Supplementary Tables 6 and 7. (E) Box plots of the correlation coefficient (R^2) values computed between Broccoli and Pepper fluorescent signals. Outcomes of statistical tests are provided in Supplementary Table 3. In D-E, each data point refers to an individual *E. coli*. In B-E, data are obtained from 3 separate cultures, each starting from a single colony.

Supplementary Figure 24: Average fluorescence intensity heatmaps of cells co-expressing two NSs (A) Average heatmap generated from cells oriented randomly along the longitudinal axis. From left, cells co-expressing NSs $A+B$, $A+B\bar{}$, $\bar{A}+B$ and $\bar{A}\bar{B}$. **(B)** Average heatmap generated from cells co-expressing NSs $A+B$ aligned such that the pole with highest Broccoli fluorescence is positioned on the left. In Fig. 2B(ii), maps were generated by aligning cells based on the pole with the highest Pepper fluorescence. These alternative maps are shown here for completeness.

Additionally, could the authors clarify whether chromatic aberration was corrected and what calibration methods were used?

Both objectives used in this study, the Plan-Apochromat 63×1.4 N.A. Oil DIC M27 (Zeiss LSM 800) and the CFI Plan Apo λ D 60×1.42 N.A. oil immersion (Nikon Ti2 Eclipse), are apochromatic and designed to minimise chromatic aberration. As such, no additional calibration or correction was deemed necessary. Two fluorescent channels were used in our experiments, one for the Pepper Aptamer (NS A) and one for the Broccoli Aptamer (NS B) and GFP. The absence of significant chromatic aberrations between these two channels is confirmed by the evidence of near-ideal co-localisation of GFP with A-type condensates, demonstrated in Fig. 3B(i) and 3E.

The main text has been revised to include this information:

“Incidentally, the strong co-localisation between GFP and A_{AP3} confirms the limited impact of potential chromatic aberrations between the red fluorescence channel (used for Pepper-labelled NSs) and the green channel (used for GFP and Broccoli-labelled NSs.)”

Extended Figure 1B suggests possible mixing of condensate types. Is there direct evidence of mixing or interaction between droplet species in vivo?

We thank the Reviewer for pointing out this matter. This point was also raised by Reviewer 1 and has been addressed above; however, we include a response here for the convenience of Reviewer 2.

As noted by the Reviewer, Extended Fig. 1B shows fluorescent signal from the Broccoli channel co-localised with A NS condensates in samples also containing soluble \bar{B} NSs. This signal may arise either from weak partitioning of \bar{B} into A condensates or from bleed-through of the Pepper (NS A) emission into the Broccoli channel. To pinpoint the origin of the effect, in the revised submission we performed image segmentation on micrographs of in vitro-expressed condensates, and present the results in Supplementary Fig. 7.

First, we quantified fluorescence in the Broccoli channel (used to image NSs B and \bar{B}) in samples containing only the condensate-forming NS A, labelled with Pepper FLAPs (Supplementary Fig. 7A). The amount of bleed-through from the Pepper channel into the Broccoli channel was minimal.

We then performed the same analysis on samples expressing A NSs together with dispersed \bar{B} NSs and, unsurprisingly, observed substantially higher signal in the Broccoli channel (Supplementary Fig. 7B). As noted in our initial submission, the signal within A condensates was higher than in the surrounding background, an effect initially ascribed to bleed-through.

In Supplementary Fig. 7C, we computed the difference, δ , between the Broccoli signal measured within A condensates and that in the surrounding background. Comparing A-only and A+ \bar{B} samples, δ is substantially higher in the latter. Two effects could contribute to this: (i) bleed-through from the Pepper dye being stronger within A-type condensates than in the surrounding solution, and (ii) a difference in \bar{B} NS concentration between the A-type condensates and the background. If (i) were the only effect—i.e. if the concentration of \bar{B} within A condensates were equal to that in the background—then similar δ values would be observed in both A and A+ \bar{B} samples. Instead, the much greater δ in A+ \bar{B} suggests that \bar{B} accumulates within A condensates, causing the increased Broccoli signal. We surmise that this accumulation arises from weak affinity between the scrambled KOs on \bar{B} and the A KOs.

Such non-specific affinity is not observed in condensate-forming B NSs. In A+B samples, Broccoli fluorescence is higher in the background than in A condensates, consistent with excluded volume effects that lower the concentration of the dispersed B phase within A condensates relative to the surrounding solution. The absence of A-B condensate adhesion further supports the lack of non-specific A-B affinity.

In vivo, we also observed \bar{B} accumulation within NS A condensates, as shown in (Fig. 2B, 2E, and Table 1), and discussed in the main text:

“The soluble NSs may indeed weakly partition within, or coat the specifically-assembled MLO, or the condensates may serve as heterogeneous nucleation sites for non-specific aggregation. Either process may be facilitated by weak affinity between the soluble and sticky NSs due to base pairing, misfolding, enzymatic degradation, or the action of native RNA-binding proteins.”

In the revised manuscript, we have incorporated this new in vitro analysis to further support partitioning of NS \bar{B} into NS A condensates.

“The greater abundance of non-specific clusters seen when non-sticky NS are co-expressed with sticky motifs, particularly for A+ \bar{B} expression, may be due to MLO-templated clustering of the otherwise soluble NSs. The soluble NSs may indeed weakly partition within, or coat the specifically-assembled MLO, or the condensates may serve as heterogeneous nucleation sites for non-specific aggregation. Either process may be facilitated by weak affinity between the soluble and sticky NSs due to base pairing, misfolding, enzymatic degradation, or the action of native RNA-binding proteins. This interpretation is supported by the evidence of weak partitioning of soluble NS \bar{B} in NS A condensates observed in vitro (Extended Data Fig. 1), and quantified in Supplementary Fig. 7 (Methods).”

We have included a summary of the discussion provided here in the caption of the new Supplementary Fig. 7.

“In *E. coli* producing A and \bar{B} , a low R^2 follows from the localised Pepper (NS A) signal in a diffused Broccoli (NS B) background. The higher R^2 value found for cells expressing $A+\bar{B}$ supports the hypothesis that \bar{B} may non-specifically partition within MLOs, consistent with observations made in vitro (Extended Data Fig. 1B, Supplementary Fig. 7).”

Supplementary Figure 7: Quantification of the Broccoli signal observed in A-type condensates. (A) Mean Broccoli fluorescence within and outside (bulk phase) NS A condensates in NS A-only samples. The Broccoli signal in the denser NS A condensates differs only slightly from that in the bulk phase, indicating that fluorescence bleed-through from the Pepper channel into the Broccoli channel is minimal. (B) Mean Broccoli fluorescence within and outside NS A condensates in $A+\bar{B}$ and $A+B$ samples. In $A+\bar{B}$ samples, Broccoli fluorescence is higher within NS A condensates than in the bulk, whereas in $A+B$ samples, fluorescence is higher in the bulk phase. The latter observation is consistent with excluded volume effects that lower the concentration of NS B within A condensates relative to the bulk phase. (C) Difference, δ , in mean Broccoli fluorescence intensity between NS A condensates and the surrounding bulk phase in A-only and $A+\bar{B}$ samples. The larger δ in $A+\bar{B}$ samples suggests accumulation of NS \bar{B} within A condensates, likely due to weak affinity between the scrambled KFs of NS \bar{B} and the KFs of NS A.

Extended Figure 10 shows droplets predominantly localised to one side of the cell when coexpressed. What might explain this asymmetric distribution? Could slight condensate mixing provide a nucleation point for B onto A?

We thank the reviewer for the insightful observation, which prompted us to analyse the localisation of A and B MLOs when co-expressed. The data shows that there is indeed an asymmetric distribution, and it is more likely for A and B MLOs to be on the same side of the cell. To estimate this asymmetry, we considered cells containing exactly one A-type and one B-type MLO, a sub-population accounting for 60% of all segmented cells. In this sub-population, 66% of the cells had both of the MLOs at one pole, while the remaining 34% had the two MLOs at opposite poles. Cells displaying a greater number of A and/or B condensates are less interesting to study, as they are bound to have A and B condensates co-localised in at least one of the poles.

The asymmetric distribution could originate from the division history of the cells, as newly formed poles are less likely to contain condensates or the nanostar-coding plasmid compared to the old poles. We plan to focus on quantifying and gaining a mechanistic insight on condensate redistribution upon division in subsequent studies. Non-specific affinity between different nanostar types could also favour condensate formation on the same pole although, as discussed in the previous point, this appears to occur primarily between soluble B NS and condensate-forming A, and not between A and B NSs.

We have made multiple changes to the manuscript based on the analysis. Firstly, Extended Data Figure 10 has been revised to include cells with condensates on opposite poles and on same poles, better reflecting the observed phenotypes. Second, we have expanded main text to discuss this observation:

“Of the cells containing one A-type and B-type MLO (60% of all cells co-expressing NSs A and B), 66% had both MLOs positioned at the same pole. This asymmetric distribution may stem from the cells’ division history, as newly formed poles are less likely to harbour condensates. Alternatively, or in addition, it may be driven by weak, non-specific affinity between A and B condensates. We also observe a clear asymmetry between the exact positioning of A and B condensates: the former are nearly always in contact with a pole, while the latter may occupy more central locations (Extended Data Fig. 2).”

Extended Data Figure 10: Zoomed in micrographs of cells co-expressing NSs A and B. Zoomed in confocal micrographs recorded in the Pepper channel (NS A, left), the Broccoli channel (NS B, centre) and merging both fluorescence channels with bright-field images (right) of *E. coli* co-expressing NS A and NS B, 1 h after IPTG induction. Scale bars, 2 μ m.

6. Temperature Dependence of Condensates. In Figure 4Ai, the uniform RNA control appears brighter than condensate-containing cells after normalisation. Since normalisation by maximum intensity varies across cells, this may be misleading; raw intensity comparisons would be more informative.

The Reviewer has raised a good suggestion to improve data visualisation. We have revised the figure such that now all images now share the same contrast scale. Intriguingly, the condensates become brighter after the temperature ramp, suggesting that heating might have increased the dye penetration into the cell. We have therefore included a comment on the increase in fluorescence in the main text:

“For both A and B NSs, condensate fluorescence intensity appears to increase after the heating and cooling ramps (Fig. 4A(i), B(i)), which may be attributed to thermally induced alterations in membrane permeability, facilitating greater intracellular accumulation of FLAP dyes.”

Fig. 4: Synthetic RNA MLOs reversibly disassemble and reassemble in *E. coli* upon temperature cycling. (A) Reversible disassembly and reassembly of A MLOs. (i) Merged epifluorescence and bright-field micrographs showing disassembly and reassembly of A MLOs in a single cell. Scale bars, 1 μm . (ii) Population-averaged maps of Pepper (NS) fluorescence as a function of temperature upon heating and subsequent cooling. (iii) Cell-population distribution of the values of skewness computed for the fluorescence-intensity distributions of Pepper signal in individual *E. coli* on heating and cooling. (iv) Percentage of single cells containing MLOs as extracted from the skewness distributions (Methods). (B) Reversible disassembly and reassembly of B MLOs. Data are presented as in panel A, using Broccoli fluorescence to image NSs. (C) Disassembly and reassembly of A_{AP3} MLOs with captured GFP. (i) Confocal micrographs of samples at 37 $^{\circ}\text{C}$ (left), after heating to 70 $^{\circ}\text{C}$ (middle), and after cooling back to 37 $^{\circ}\text{C}$ (right). Top: Pepper (NS) fluorescence; middle: GFP fluorescence; bottom: merged Pepper fluorescence, GFP fluorescence, and bright-field. Scale bars, 10 μm . (ii) Population-averaged maps of Pepper (top) and GFP (bottom) fluorescence at the temperatures shown in (i). In A–C, fluorescence maps and quantitative analyses are based on *E. coli* from 3 independent cultures, each starting from a single colony.

What is the in vitro melting temperature of the condensates?

We thank the Reviewer for the interesting question. In our initially submitted manuscript, we referenced measurements made by Fabrini et al [2] of the melting temperature of similar NS designs. Here, we have implemented design changes compared to Fabrini et al., which may impact the melting temperature. Hence, in the revised submission, we included new measurements to determine the melting temperatures of in vitro NS A and NS B condensates (Methods). Following Ref. [33], the melting temperature was determined using a thermal microscope stage, monitoring the samples via epifluorescence imaging during a heating ramp, followed by a cooling ramp, as detailed in the updated Methods section. We then analysed the epifluorescence frames by computing the standard deviation of the pixel values in the relevant fluorescent channel, σ , and the mean pixel intensity I . The quantity σ/I , plotted against T , displays a sharp transition when condensates form or melt, which can be used to determine the melting (on heating, T_m) and condensate-formation (on cooling, T_f) temperatures [33]. Results are collated in new Supplementary Fig. 36, with Supplementary Figs. 37 and 38 containing relevant epifluorescence snapshots. The new figures are included at the end of this document for quick reference. As expected, and previously observed with similar DNA condensates [33], for both A and B T_m -values are slightly higher compared to T_f . This is due to condensate assembly timescales being slower than those of condensate melting, and possible small delays in heat propagation through the sample cell. We additionally note that T_m values are slightly higher for NS B compared to NS A, 40.9 °C vs 39.9 °C. This trend is consistent with the observations of Fabrini et al. [2], with the small absolute differences likely due to design changes and differences in the IVT buffer and assembly protocol used (that may impact NS concentration and KL-KL affinity). More remarkably, as already noted in the initial submission, the in-vitro melting temperatures are substantially lower compared to those measured in vivo (Fig. 4), likely due to a combination of different ionic conditions, and higher molecular crowding.

The main text was revised to describe and discuss the new data:

“Since the NSs self-assemble through base-pairing interactions, we anticipated that the MLOs would dissolve at sufficiently high temperature, as observed in vitro (Supplementary Figs. 36, 37, 38). Consistent with this expectation, heating *E. coli* containing A-MLOs led to their dissolution at 61 °C.”

“The difference in melting temperature between the two constructs is unlikely to result from different KL affinity, having observed that NSs with KL A disassemble at slightly lower temperature (39.9 °C) compared to those with KL B in vitro (40.9 °C, Supplementary Figs. 36, 37, 38).”

“For both A and B NSs, condensate fluorescence intensity appears to increase after the heating and cooling ramps (Fig. 4A(i), B(i)), which may be attributed to thermally induced alterations in membrane permeability, facilitating greater intracellular

accumulation of FLAP dyes. We further note that melting temperatures measured in vivo are substantially higher than those we observe in vitro, where the MLOs melted at 39.9-40.9 °C (Supplementary Fig. 36, 37, 38, 43). This deviation likely originates from intracellular molecular crowding promoting condensation [34] or difference in ionic conditions between the in vitro transcription buffer and the cytoplasm.”

Supplementary Figure 36: Thermal transition curves of A and B condensates expressed in vitro. The thermal transition curves of NS A condensates are shown in orange, and those of NS B condensates in cyan. Solid curves corresponding to the heating ramp were used to estimate the melting temperatures (T_m), while dashed curves corresponding to the cooling ramp were used to estimate the condensate-formation temperatures (T_f). Data show the ratio between the standard deviation of the pixel values (σ) and the mean pixel intensity (I) of the epifluorescence micrographs. The quantity σ/I , plotted against T displays a sharp transition when condensates melt or form, which can be used to estimate T_m and T_f . These were computed from the intersections of the relevant curve with a threshold value in σ/I , defined as the high-temperature plateau of the data (mean computed for $T > 41^\circ\text{C}$) augmented by 0.05 units (Methods). For both A and B, T_m values (39.9°C and 40.9°C , respectively) are slightly higher than their corresponding T_f values (36.9°C and 38.6°C , respectively), reflecting the slower timescales of condensate assembly compared to condensate melting. Notably, the calculated T_m and T_f values are substantially lower than those observed in vivo, likely due to a combination of different ionic conditions and molecular crowding. Samples were initially incubated at 30°C for 15 hours, after which the temperature was gradually changed from 30°C to 50°C and back to 30°C , with 1°C steps introduced every 5 minutes.

Supplementary Figure 37: Epifluorescence micrographs of NS A (left) and NS B (right) condensate melting upon increasing temperature. After the initial 15 hour incubation at 30 °C, the temperature was gradually increased from 30 °C to 50 °C in 1 °C steps introduced every 5 minutes, leading to NS condensate melting. Scale bars, 100 μ m.

Supplementary Figure 38: Epifluorescence micrographs of NS A (left) and NS B (right) condensate formation upon cooling. The temperature was gradually decreased from 50 °C to 30 °C in 1 °C steps introduced every 5 minutes, leading to NS condensate-formation. Scale bars, 100 μm .

Does melting temperature correlate with cellular expression level or fluorescence intensity? Given theoretical predictions, higher nanostar concentration should increase the melting temperature until the critical point, after which it declines. Exploring this relationship experimentally would enhance the thermodynamic analysis.

We thank the Reviewer for the insightful suggestion. To address this, we analyzed the temperature-dependent data for the skewness of the fluorescence-intensity distributions recorded in single cells, which we used to determine the population distributions in Fig. 4A(iii) and B(iii). This produced skewness-temperature curves that could, potentially, be used to extract condensate T_m for single cells. For each curve that showed a skewness above 0.25 at the initial temperature of 37 °C (indicating the presence of condensates), T_m was estimated as the inflection point of a sigmoid fit (Supplementary Fig. 42 Left, included below). We then produced a scatter plot of T_m values against the integrated fluorescence at 37 °C, for each cell (Supplementary Fig. 42 Right). Linear fits of the scatter plots yielded a very low coefficient of determination ($R^2 < 0.2$), indicating no correlation between fluorescence intensity and melting temperature.

However, we point out that several sources of error or uncertainty are likely to impact these data. First, for NS A, skewness-temperature curves did not reach their high-temperature plateau at the maximum temperature we probed (61 °C), making the sigmoid fit imprecise. Furthermore, for both NS types, the reliability of the sigmoid fit is limited by the number of temperatures probed and the intrinsic noise of the measurements. We have included a summary of the discussion in the caption of the new Supplementary Fig. 42, included below for quick reference:

The main text was revised to discuss the new data:

“Single-cell skewness-temperature curves were used to estimate the melting temperature for each cell. Linear regression between the melting temperature and cellular fluorescence gave low coefficients of determination ($R^2 < 0.2$), indicating no detectable correlation between NS concentration and melting temperature (Supplementary Fig. 42).”

Supplementary Figure 42: Analysis of the relationship between fluorescence intensity and melting temperature. Each row contains data from one biological replicate of *E. coli* expressing NS A (top) or NS B (bottom). The left panels show single-cell skewness as a function of temperature, with the melting temperature determined by fitting a sigmoid curve (indicated by dots). The right panels display the relationship between the melting temperature and the integrated cellular fluorescence at the starting temperature of 37 °C. Linear fits were used to calculate the coefficients of determination (R^2) for NS A ($R^2 = 0.00$) and NS B ($R^2 = 0.18$). We note that, for NS A, skewness-temperature curves did not reach their high-temperature plateau at the maximum temperature we probed (61 °C), making the sigmoid fit imprecise. Furthermore, for both NS types, the reliability of the sigmoid fit is limited by the number of temperatures probed and the intrinsic noise of the measurements.

Code availability – readme file missing

The readme file has been updated.

References

- [1] Wesselhoeft, R. A., Kowalski, P. S. & Anderson, D. G. Engineering circular rna for potent and stable translation in eukaryotic cells. *Nature Communications* **9**, 1–10 (2018).
- [2] Fabrini, G. *et al.* Co-transcriptional production of programmable RNA condensates and synthetic organelles. *Nature Nanotechnology* *2024* 1665–1673 (2024).
- [3] Workman, R. E. *et al.* Nanopore native RNA sequencing of a human poly(A) transcriptome. *Nature Methods* **16**, 1297–1305 (2019).
- [4] Gan, J. *et al.* Structural insight into the mechanism of double-stranded rna processing by ribonuclease iii. *Cell* **124**, 355–366 (2006).
- [5] Hurwitz, C. & Rosano, C. L. The intracellular concentration of bound and unbound magnesium ions in escherichia coli. *Journal of Biological Chemistry* **242**, 3719–3722 (1967).
- [6] Radchenko, M. V. *et al.* Potassium/proton antiport system of escherichia coli. *Journal of Biological Chemistry* **281**, 19822–19829 (2006).
- [7] Mitsuzawa, S., Deguchi, S. & Horikoshi, K. Cell structure degradation in *Escherichia coli* and *Thermococcus* sp. strain Tc-1-95 associated with thermal death resulting from brief heat treatment. *FEMS Microbiology Letters* **260**, 100–105 (2006).
- [8] Cooper, S. T. & McNeil, P. L. Membrane Repair: Mechanisms and Pathophysiology. *Physiological Reviews* **95**, 1205–1240 (2015).
- [9] Vauclore, P. *et al.* Stress-induced nucleoid remodeling in *Deinococcus radiodurans* is associated with major changes in Heat Unstable (HU) protein dynamics. *Nucleic Acids Research* **52**, 6406–6423 (2024).
- [10] Vo, K. C., Sakamoto, J. J., Furuta, M. & Tsuchido, T. The impact of heat treatment on e. coli cell physiology in rich and minimal media considering oxidative secondary stress. *Journal of Applied Microbiology* **135** (2024).
- [11] Stringer SC., G. S. & MW., P. Thermal inactivation of Escherichia coli O157:H7. *Journal of Applied Microbiology Symposium Supplement* **88**, 798–898 (2000). Symposium Supplement.
- [12] Bromberg, R., George, S. M. & Peck, M. W. Oxygen sensitivity of heated cells of escherichia coli o157:h7. *Journal of Applied Microbiology* **85**, 231–237 (1998).
- [13] Geary, C., Grossi, G., McRae, E. K., Rothmund, P. W. & Andersen, E. S. RNA origami design tools enable cotranscriptional folding of kilobase-sized

- nanoscaffolds. *Nature Chemistry* **13**, 549–558 (2021).
- [14] Stewart, J. M. *et al.* Modular RNA motifs for orthogonal phase separated compartments. *Nature Communications* **15**, 6244 (2024).
- [15] Udono, H. *et al.* Programmable Computational RNA Droplets Assembled via Kissing-Loop Interaction. *ACS Nano* **18**, 15477–15486 (2024).
- [16] Li, M. *et al.* In vivo production of RNA nanostructures via programmed folding of single-stranded RNAs. *Nature Communications* **9**, 2196 (2018).
- [17] Xie, M., Chen, W., de Roy, M. V. & Walther, A. Constructing synthetic nuclear architectures via transcriptional condensates in a dna protonucleus. *Nature Communications* *2025 16:1* **16**, 1–12 (2025).
- [18] Gao, D. *et al.* Controlling the size and adhesion of dna droplets using surface-enriched dna molecules. *Soft Matter* **20**, 1275–1281 (2024).
- [19] Benza, V. G. *et al.* Physical descriptions of the bacterial nucleoid at large scales, and their biological implications. *Reports on Progress in Physics* **75**, 076602 (2012).
- [20] Kleckner, N. *et al.* The bacterial nucleoid: nature, dynamics and sister segregation. *Current Opinion in Microbiology* **22**, 127–137 (2014).
- [21] Pelletier, J. *et al.* Physical manipulation of the escherichia coli chromosome reveals its soft nature. *Proceedings of the National Academy of Sciences of the United States of America* **109**, E2649–E2656 (2012).
- [22] Zadeh, J. N. *et al.* NUPACK: Analysis and design of nucleic acid systems. *Journal of Computational Chemistry* **32**, 170–173 (2011).
- [23] Coquel, A. S. *et al.* Localization of protein aggregation in escherichia coli is governed by diffusion and nucleoid macromolecular crowding effect. *PLOS Computational Biology* **9**, e1003038 (2013).
- [24] Wlodarski, M. *et al.* Cytosolic crowding drives the dynamics of both genome and cytosol in escherichia coli challenged with sub-lethal antibiotic treatments. *iScience* **23**, 101560 (2020).
- [25] Coquel, A. S. *et al.* Localization of Protein Aggregation in Escherichia coli Is Governed by Diffusion and Nucleoid Macromolecular Crowding Effect. *PLOS Computational Biology* **9**, e1003038 (2013).
- [26] Jin, X. *et al.* Membraneless organelles formed by liquid-liquid phase separation increase bacterial fitness. *Science Advances* **7** (2021).

- [27] Nandana, V. & Schrader, J. M. Roles of liquid-liquid phase separation in bacterial RNA metabolism. *Current opinion in microbiology* **61**, 91 (2021).
- [28] Nguyen, M., Pothoulakis, G. & Andersen, E. *Nucleic Acids Research* **50**, 7176–7189 (2022).
- [29] Nguyen, M., Pothoulakis, G. & Andersen, E. Synthetic Translational Regulation by Protein-Binding RNA Origami Scaffolds. *ACS Synthetic Biology* **11**, 1710–1718 (2022).
- [30] Delebecque, C., Lindner, A., Silver, P. & Aldaye, F. Organization of intracellular reactions with rationally designed RNA assemblies. *Science* **333**, 470–474 (2011).
- [31] Sachdeva, G., Garg, A., Godding, D., Way, J. C. & Silver, P. A. *Nucleic Acids Research* **42**, 9493–9503 (2014).
- [32] Calvopina-Chavez, D. G., Gardner, M. A. & Griffiths, J. S. Engineering efficient termination of bacteriophage T7 RNA polymerase transcription. *G3 Genes—Genomes—Genetics* **12**, jkac070 (2022).
- [33] Tanase, D. A. *et al.* Internal phase separation in synthetic dna condensates. *Advanced Science* **12**, e06275 (2025).
- [34] Singh, A., Maity, A. & Singh, N. Structure and Dynamics of dsDNA in Cell-like Environments. *Entropy* **24**, 1587 (2022).

Response to reviewers' comments (Round 2)

Points raised by the Reviewers are highlighted in blue, while our responses are presented in black.

REVIEWERS' COMMENTS

Reviewer #1 (Remarks to the Author):

Great effort during the revisions. This is a great study and is ready for publication.

We sincerely thank the Reviewer for their positive assessment and for the constructive suggestions provided during the earlier stages, which helped us strengthen the manuscript.

Reviewer #2 (Remarks to the Author):

I appreciate the authors' detailed and thorough response, which included substantial new experimental work, data analysis, and writing that significantly enhance the quality of the manuscript. My concerns have been addressed, and I support publication.

We are grateful to the Reviewer for their careful evaluation and for encouraging us to expand and refine the manuscript. We are pleased that the additional experiments and analyses have resolved their concerns and that they now support publication.